# From Word to Sentence: A Large-Scale Multi-Instance Dataset for Open-Set Aerial Detection

## Abstract

In recent years, language-guided open-world aerial object detection has gained significant attention due to its better alignment with real-world application needs. However, due to limited datasets, most existing language-guided methods primarily focus on vocabulary, which fails to meet the demands of more fine-grained open-world detection. To address this limitation, we propose constructing a large-scale language-guided open-set aerial detection dataset, encompassing three levels of language guidance: from words to phrases, and ultimately to sentences. Centered around an open-source large vision-language model and integrating image-operation-based preprocessing with BERT-based postprocessing, we present the **OS-W2S Label Engine**, an automatic annotation pipeline capable of handling diverse scene annotations for aerial images. Using this label engine, we expand existing aerial detection datasets with rich textual annotations and construct a novel benchmark dataset, called Multi-instance Open-set Aerial Dataset (**MI-OAD**), addressing the limitations of current remote sensing grounding data and enabling effective open-set aerial detection. Specifically, MI-OAD contains 163,023 images and 2 million image-caption pairs, with multiple instances per caption, approximately 40 times larger than the comparable datasets. We also employ state-of-the-art open-set methods from the natural image domain, trained on our proposed dataset, to validate the model's open-set detection capabilities. For instance, when trained on our dataset, Grounding DINO achieves improvements of 31.1 $AP_{50}$ and 34.7 Recall@10 for sentence inputs under zero-shot transfer conditions. Both the dataset and the Label Engine will be made publicly available.

## 1 Introduction

Object detection is indispensable for accurately identifying and localizing objects of interest in aerial imagery [5]. It plays a crucial role in various applications, such as environmental monitoring, urban planning, and rescue operations [1, 25, 34]. Most existing aerial detectors primarily focus on addressing the inherent challenges of aerial images and are limited to fixed categories and scenarios, which defines them as closed-set detectors. However, as the demand for more versatile applications increases, closed-set detectors become inadequate for meeting real-world requirements.

Recently, language-guided open-world object detection has garnered significant attention due to its alignment with real-world application requirements. Several studies [12, 16, 24, 30] have explored open-vocabulary aerial detection. CastDet [12] employs a multi-teacher architecture that leverages the superior image-text alignment capabilities inherited from pre-trained VLMs. OVA-Det[24] proposes a lightly open-vocabulary aerial detector that adopts a text-guided strategy to further enhance image-text alignment. These methods are constrained by the limited category diversity in aerial detection, which provides minimal semantic information. Besides, there is an approach that addresses this

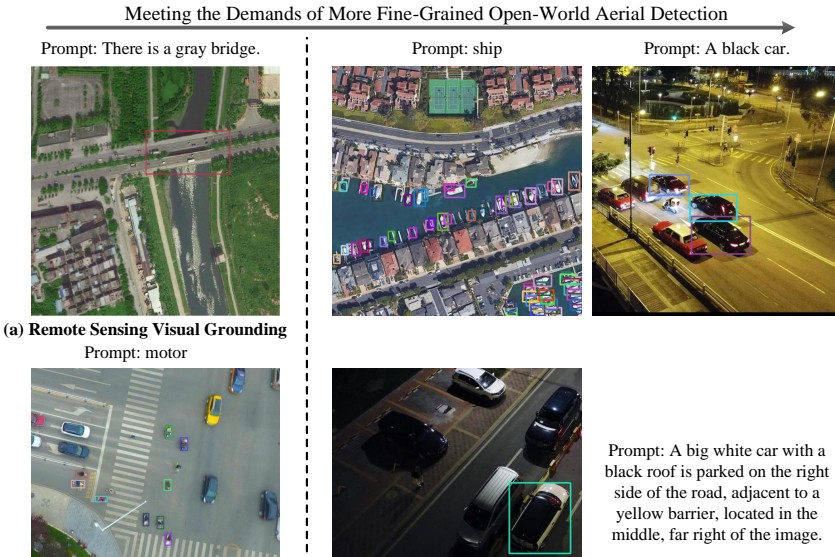

Meeting the Demands of More Fine-Grained Open-World Aerial Detection

Prompt: There is a gray bridge.   Prompt: ship   Prompt: A black car.

**(a) Remote Sensing Visual Grounding**

Prompt: motor

Prompt: A big white car with a black roof is parked on the right side of the road, adjacent to a yellow barrier, located in the middle, far right of the image.

**(b) Open-Vocabulary Aerial Detection**   **(c) Open-Set Aerial Detection (Data from MI-OAD)**

Figure 1: (a) Remote-sensing visual grounding focuses on precise object localization, corresponding to a single instance only, and lacks caption diversity due to its reliance on template-generated captions. (b) Open-vocabulary aerial detection is constrained by a limited number of aerial categories, which have only minimal semantic richness. (c) Open-set aerial detection supports multi-level descriptive detection, ranging from words to phrases, and ultimately to richly detailed sentences.

limitation from the dataset perspective: LAE-DINO [16] employs VLMs to expand the number of detectable categories, aiming to increase category diversity and enrich the semantic content of the detection text. Although these methods effectively equip models with open-vocabulary capabilities to overcome the category limitations of traditional aerial detectors, their practical applicability remains constrained by the weak semantic representation of categories, which are typically represented by a single word. In other words, there is still significant room for optimization.

Compared to the aerial domain, open-set object detection in natural scenes has achieved advancement significantly [13, 21]. We note that this is primarily due to the abundance of grounding data available for natural scenes. For instance, Grounding DINOv1.5 provides robust open-set detection capability by training on over 20 million grounding samples. In contrast, aerial grounding data is scarce. Only a few attempts [10, 23, 31] have been made to construct remote sensing visual grounding (RSVG) datasets by annotating detection data with captions, yet these datasets suffer from several limitations: *1) Lack of scene diversity*: Dataset construction is restricted to images containing no more than five objects of the same category, to ensure the correct correspondence between captions and instances, resulting in only simplistic scenes. *2) Limited caption diversity*: Captions are generated using fixed templates, restricting their variability. *3) Single-instance annotation*: Current RSVG datasets solely emphasize precise localization, where each image-caption pair corresponds to a single instance. Including cases where a vague caption corresponds to multiple instances within an image is critical for practical applications. *4) Limited Dataset Scale*: The largest available dataset comprises only 25,452 images and 48,952 image-caption pairs. These limitations make existing datasets inadequate for open-set aerial object detection, and the scarcity of large-scale, semantically rich grounding data remains a major bottleneck in advancing the field.

To bridge this gap, in this paper, we aim to lay the data foundation for open-set aerial object detection. Specifically, we propose the *OS-W2S* Label Engine, an automatic annotation pipeline capable of handling diverse scene annotations for aerial images. It is based on an open-source vision-language model, image-operate-based preprocessing, and BERT-based postprocessing. Using this label engine, we construct a novel large-scale benchmark dataset, called MI-OAD, to overcome the limitations of current RSVG data.

Key aspects include: *1) Scene Diversity:* As depicted in Fig. 2, we introduced pre-processing steps (e.g., extracting foreground and instance regions) and post-processing steps (e.g., matching caption-instance associations for each image) both before and after interactions with the VLM. This design enables the pipeline to effectively handle various scenarios aerial images and ensure label quality. *2) Caption Diversity:* Leveraging the robust vision-language capabilities of the VLM, we generate captions with varying levels of detail for each instance based on its attributes, thereby ensuring caption diversity. *3) Multi-instance annotation:* We aim to match varying numbers of instances to each caption based on its descriptive details during the post-processing steps. This process enables the generated data to meet diverse requirements in practical applications, accommodating both precise and approximated localization. *4) Dataset Scale:* Using this label engine, we expanded eight widely used aerial detection datasets, yielding 163,023 images and 2 million image-caption pairs, which is 40 times larger than those available in existing RS grounding datasets.

In summary, our contributions are three-fold: (1)We introduce the OS-W2S Label Engine, an automatic annotation pipeline that lays the data foundation for open-set aerial object detection and can be executed on a single workstation equipped with eight RTX4090 GPUs. (2)Using this engine, we present MI-OAD, the first benchmark for open-set aerial object detection, encompassing 2 million image–caption pairs with multiple instances per caption, annotated at the word-, phrase-, and sentence-level. (3)We show that training mainstream open-set detectors, originally designed for natural images, on MI-OAD leads to significant gains in open-set aerial object detection performance.

## 2 Related Work

### 2.1 Open-set Object Detection

Open-set object detection, which refers to detecting objects based on arbitrary textual inputs, demonstrates significant potential due to its close alignment with real-world application needs. Several studies [3, 11, 13, 20, 29, 32] have demonstrated the feasibility of open-set object detection in the natural image domain. GLIP [11] established a foundation for open-set detection by integrating object detection and grounding tasks. Building on this, models such as YOLO-World [3] and the Grounding DINO series [13, 20, 21] have made significant progress. Notably, Grounding DINO v1.5, trained on over 20 million images with grounding annotations, demonstrates exceptional open-set detection performance, underscoring the crucial role of large-scale grounding data.

Compared to the natural image domain, the development of open-set aerial object detection has lagged behind, primarily due to a lack of sufficient grounding data in aerial contexts. To bridge this gap, this paper aims to establish a data foundation for open-set aerial object detection.

### 2.2 Object Detection in Aerial Imagery

Aerial object detection can be bordely divide into two types: closed-set aerial detection and open-vocabulary aerial detection.

Closed-set aerial detection refers to predicting bounding boxes and corresponding categories for objects that have been seen during training. Several studies [4, 6, 8, 14, 28] have primarily focused on addressing the inherent challenges of RS images. For instance, models such as UFPMP-Det [6], ClustDet [28], and DMNet [8] employ a coarse-to-fine two-stage detection architecture to mitigate significant background interference and effectively detect tiny, densely distributed objects. However, these models are constrained by predefined training categories, making them suitable only for specific scenarios in real-world applications.

Open-vocabulary aerial detection marks a step towards meeting the demands of open-world aerial detection. It seeks to eliminate the category limitations inherent in closed-set detection by establishing a relationship between image features and category embeddings, rather than simply linking image features to category indices. Models such as CastDet [12], DescReg [30], and OVA-Det [24] leverage the superior image-text alignment capabilities inherited from pre-trained Visual Language Models (VLMs) to enable open-vocabulary aerial detection capabilities. However, the performance of these models is constrained by a limited number of categories in aerial detection. Additionally, LAE-DINO [16] aim of addressing this limitation from a dataset perspective. It employs VLMs to expand the detection category set, thereby increasing category diversity and enriching the semantic content of the detection text.

Despite these advancements, current research in open-vocabulary aerial detection remains limited at the vocabulary level—relying on only a few words that offer scant semantic information. Compared with the natural image domain, open-set object detection in aerial images still has significant room for exploration and improvement.

## 2.3 Visual Grounding in Aerial Imagery

Visual grounding in remote sensing (RSVG) aims to locate objects based on natural language descriptions. Compared to close-set object detection, which relies on fixed category labels, RSVG can process arbitrary descriptions to identify corresponding bounding boxes, offering greater flexibility and suitability for practical applications [10]. However, this flexibility also introduces additional complexity to the RSVG task. Currently, RSVG remains in its early stages of development, with only three publicly available datasets: RSVG-H [23], DIOR-RSVG [31], and OPT-RSVG [10]. Among these, RSVG-H comprises 4,239 RS images paired with 7,933 textual descriptions, each providing precise geographic distances (e.g., "Find a ground track field, located approximately 295 meters southeast of a baseball field."). DIOR-RSVG, based on the DIOR dataset [9], makes use of tools such as HSV and OpenCV to extract instance attributes (e.g., geometric shapes and colors) and employs predefined templates to generate 38,320 image-caption pairs. Meanwhile, OPT-RSVG further enriches RSVG scenarios by combining three detection datasets (DIOR, HRRSD [33], and SPCD [2]) and follows the annotation process in [31] to produce 25,452 RS images with 48,952 image-caption pairs.

Nevertheless, compared to the abundance of grounding data for natural images, the number of available aerial grounding data is extremely limited. This poses a significant barrier for data-driven open-set detection tasks. We observe that this issue stems from the inherent challenges in annotating aerial images, which often contain predominantly small objects and substantial background interference. Moreover, the captions in existing grounding datasets are typically generated through fixed templates, with each image-caption pair corresponding to a single instance annotation.

To address these limitations and lay the data foundation for open-set aerial object detection, this paper proposes the OS-W2S label engine and constructs MI-OAD, a large-scale benchmark dataset for open-set aerial detection tasks.

# 3 Dataset Construction

## 3.1 Motivation

In the aerial detection domain, current research primarily focuses on open-vocabulary detection, aiming to eliminate the limitations imposed by predefined categories. Although these studies have made notable progress, they remain confined to the vocabulary level, which provides only minimal semantic information and consequently limits their applicability. Developing open-set aerial detection is imperative to enable more flexible detection, thereby meeting the rapidly growing demands of fine-grained, open-world aerial detection. We observe that open-set detection in natural images has advanced significantly more than in the aerial detection domain. This disparity is primarily due to the extreme scarcity of aerial grounding data compared to that available for natural images.

To fill this gap, we propose OS-W2S Label Engine, an automatic annotation pipeline capable of handling diverse scene annotations for aerial images, and construct MI-OAD, a large-scale benchmark dataset for open-set aerial object detection tasks, thereby laying a robust data foundation for future research in this area.

## 3.2 Design of OS-W2S Label Engine

As shown in Fig. 2, the OS-W2S Label Engine consists of the following four components:

**Data Collection.** We collected eight representative aerial detection datasets [7, 9, 17, 22, 26, 27, 33, 35], ensuring diverse scenes due to variations in capturing heights and equipment (e.g., satellites and drones) across different datasets. Due to inconsistencies in image resolution and annotation formats, we standardized the resolution by cropping high-resolution images and aligning annotation formats. These processing steps, combined with annotations of instance categories and coordinates inherent to detection tasks, establish a robust foundation for the subsequent annotation pipeline.

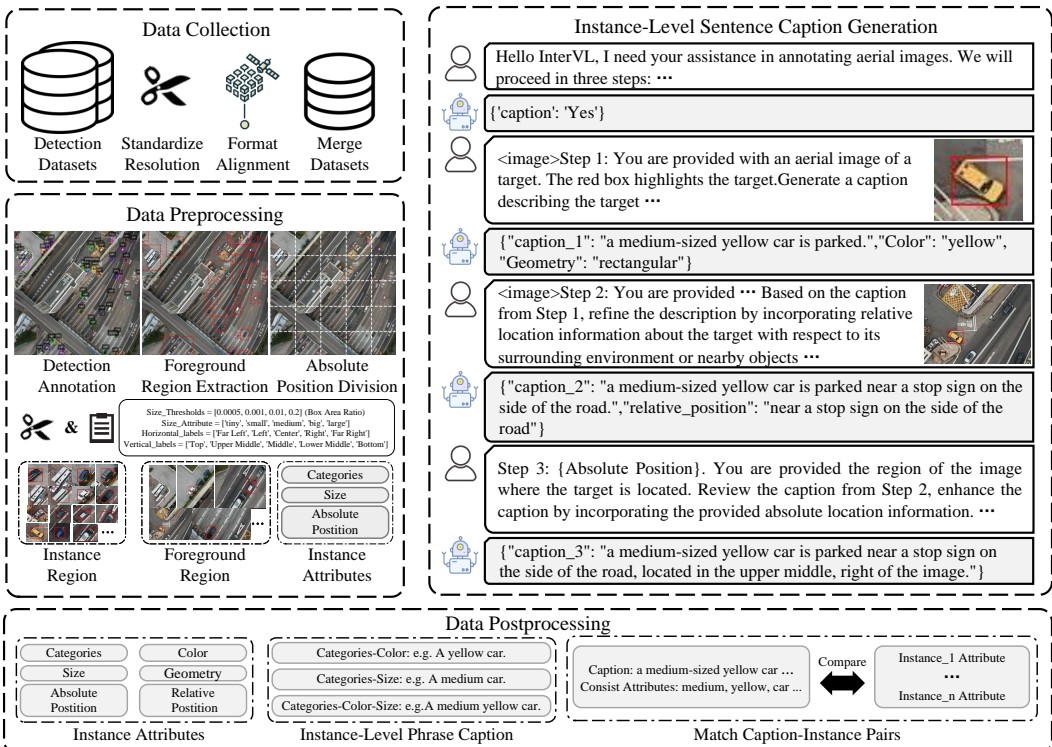

Figure 2: The pipeline of the proposed OS-W2S Label Engine. The labeling process includes four major components: Data Collection, Data Preprocessing, Instance-Level Sentence Caption Generation, and Data Postprocessing. Each aerial image undergoes a comprehensive annotation process involving attribute extraction, caption generation with varying detail levels using a VLM, and precise matching of caption-instance associations based on attribute similarity.

**Data Preprocessing.** Data preprocessing aims to simplify complex aerial images, enabling VLMs to effectively focus on relevant regions. Specifically, we process images to extract three critical components: instance regions, foreground regions, and partial instance attributes. (1) Instance regions: These are easily obtained by cropping sub-images based on the coordinates provided in the detection annotations. (2) Foreground regions: Given the dense distribution of instances and the large proportion of background in aerial imagery, we apply a foreground-extraction algorithm to isolate object regions. Specifically, we compute the maximum enclosing rectangle of the object bounding boxes to isolate multiple object clusters within each image. (3) Partial Instance Attributes: Inspired by previous approaches [15, 31], we leverage instance attributes as components to generate diverse captions. We focus on six primary attributes: category, size, color, geometric shape, relative position, and absolute position. While the category is predefined, size and absolute position attributes are determined based on manual rules due to their inherent subjective nature and spatial complexity. Specifically, size attributes are classified according to predefined thresholds, and absolute positions are categorized into 25 labeled regions (e.g., Left-Top, Far Right-Bottom). The remaining attributes are dynamically generated by the VLM based on image content during the annotation process.

**Instance-Level Sentence Caption Generation.** This step aims to interact with the VLM to generate three sentence captions with varying levels of detail and additional instance attributes for each instance. To achieve an optimal balance between annotation fidelity and computational efficiency, we employ the InternVL-2.5-38B-AWQ model, which can be executed on a single workstation equipped with eight RTX4090 GPUs. This benefits from the proposed OS-W2S Label engine, which enables high-quality caption annotation to be acquired without dependence on excessively large-scale models. The interaction with the VLM for each instance can be structured into four rounds: (1) Introduction of the overall annotation workflow to the VLM. (2) Providing the instance-specific region image along with known attributes such as instance category and size, prompting the VLM to infer additional attributes (color, geometric) and subsequently generate an initial self-descriptive

caption. (3) Presentation of the foreground region image corresponding to the instance, enabling the VLM to extract the relative positional attribute based on the surrounding context and extend the previous caption with the relative positional attribute. (4) Provision of the absolute position attribute to the VLM, prompting it to integrate this information into the existing caption, thus generating a comprehensive caption reflecting the absolute spatial context. To ensure consistent and precise VLM outputs, each interaction is regulated through structured JSON templates. Consequently, each instance is annotated with three distinct sentence captions with different levels of descriptive detail, supplemented by a set of six attributes.

**Data Postprocessing.** Based on the attributes obtained from previous steps, we generate three phrase-level captions per instance using combinations of category, color, and size attributes, resulting in six unique captions per instance. However, due to instance similarities, captions with fewer attributes often correspond to multiple instances. Leveraging attribute-based captions and the recorded attribute information for each instance, we effectively establish caption-instance associations by comparing the attribute similarity between captions and instances. The attribute similarity is computed using Sentence-BERT [19].

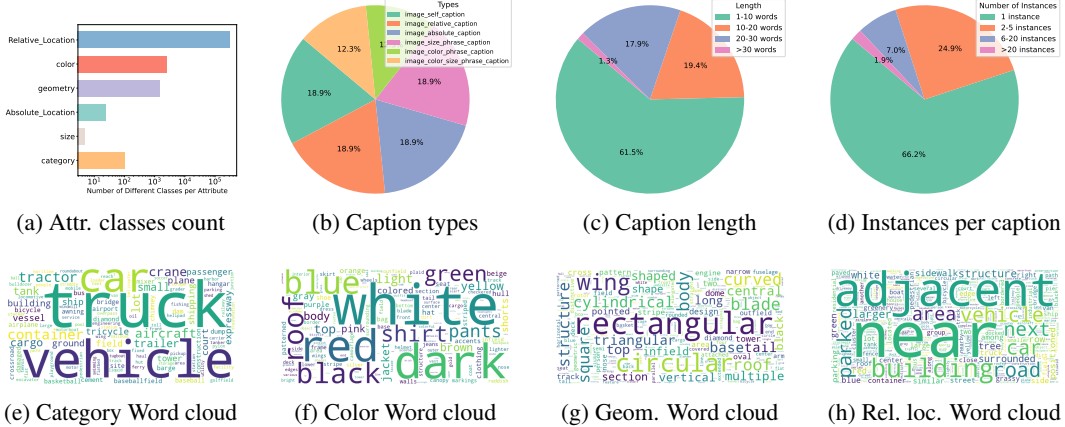

(a) Attr. classes count     (b) Caption types     (c) Caption length     (d) Instances per caption

(e) Category Word cloud     (f) Color Word cloud     (g) Geom. Word cloud     (h) Rel. loc. Word cloud

Figure 3: Statistical analysis and visualization of the MI-OAD dataset. (a) The number of distinct expressions per attribute, highlighting attribute diversity. (b) Distribution of caption types, emphasizing that the caption types are evenly distributed. (c) Distribution of caption lengths, reflecting semantic richness. (d) Distribution of instances per caption, indicating that captions correspond to both single-instance and multi-instance cases. (e) Word cloud visualization of categories directly sourced from the collected detection datasets. (f)-(h) Word cloud visualizations illustrating diverse semantic expressions for color, geometry, and relative position attributes generated by the VLM.

### 3.3 MI-OAD Dataset

Using the OS-W2S Label Engine, we created a large-scale, multi-instance dataset for open-set aerial object detection. This dataset comprises 163,023 images and 2 million image-caption pairs, encompassing three levels of language guidance: vocabulary-level, phrase-level, and sentence-level. The average caption length is 10.61 words, providing rich semantic information. Benefiting from the design of the OS-W2S Label Engine, the MI-OAD dataset effectively addresses the limitations of the existing RSVG dataset and establishes the first benchmark dataset for open-set aerial object detection.

**Scene Diversity:** We made two efforts to ensure scene diversity. First, we collected data from eight detection datasets, which include images taken from various altitudes and viewpoints using drones and satellites. Second, we generated multiple types of captions and performed both data preprocessing and postprocessing to ensure the quality of captions for complex scenes. As a result, there is no need to filter out complex scenes based on an upper limit on instance count.

**Caption Diversity:** Each caption is generated based on the attributes of instances. To ensure comprehensive coverage, we defined three sentence caption types and three phrase caption types, each varying in detail based on attribute combinations. The sentence captions provide detailed instance descriptions suitable for precise localization. Specifically, self sentence captions describe the category, size, color, and geometric attributes of instances. By adding relative positional information,

we obtain relative sentence captions, and by incorporating absolute positional information, we form absolute sentence captions. Additionally, three types of phrase captions constructed from combinations of category, color, and size attributes were created to support approximate localization.

Fig.3b illustrates the distribution of caption types, highlighting that after applying the sampling strategy described in Section4.1, the caption types are evenly distributed. Fig.3a presents the number of distinct expressions for each attribute, highlighting the rich diversity in attributes (relative location, color, and geometry) generated by the VLM. To visually demonstrate the quality of these VLM-generated attributes, we conducted a word cloud analysis as shown in Fig.3(f)-(h). Notably, the geometry attribute extends beyond basic shapes to include descriptive components (e.g., "a cylindrical tower with three blades"). Furthermore, we analyzed the distribution of caption lengths to illustrate the richness of descriptions, as depicted in Fig.3c. Collectively, these analyses underscore the caption diversity within our dataset.

**Multi-instance Annotation:** To better align with real-world applications requiring both precise and approximate localization, each caption corresponds to all relevant instances in the image matching the description, encompassing both single-instance and multi-instance cases. We construct caption-instance associations by comparing the attributes of captions and instances. As shown in Fig. 3d, 66.2% of captions correspond to a single instance, demonstrating that the generated captions effectively support precise localization even in complex scenes. The remaining captions, which involve multiple instances, fulfill the requirements for approximated localization.

**Dataset Scale:** The OS-W2S Label Engine is capable of generating high-quality caption annotations for each instance, and the aerial detection dataset contains numerous instance annotations. These conditions enable us to establish a large-scale dataset for open-set aerial detection. Finally, we constructed the MI-OAD dataset, which contains 163,023 images and 2 million image-caption pairs, making it 40 times larger than the existing RSVG dataset.

### 3.4 Quality Control Analysis

To guarantee the reliability of the captions generated by the *OS-W2S Label Engine*, we employ a three-tier quality-assurance pipeline:

- Authoritative data sources. We start from widely used aerial detection datasets whose bounding boxes and category labels have been manually verified. These well-curated sources let us inherit precise instance locations and trustworthy class information, forming a solid basis for caption generation.

- Rule-based constraints. For every instance we extract six attributes. *Category*, *size*, and *absolute location* are deterministically derived from the detection annotations. *Color* and *geometric shape* are inferred by the VLM that receives an instance-centered crop, ensuring the model attends exclusively to the target. The *relative position* attribute is obtained by supplying the VLM with a foreground region corresponding to the instance. This targeted zoom-in operation explicitly guides the VLM's attention and thus improves caption quality. We further enforce syntactic correctness through regular-expression filtering, ultimately producing three sentence-level and three phrase-level captions per instance—each with high linguistic quality.

- Two-stage manual verification. *Stage 1:* We randomly sampled 1,000 images and asked five senior experts to assess each instance and its corresponding caption. 95% were deemed correct, and the remaining discrepancies were mainly color mismatches caused by illumination changes or motion blur. *Stage 2:* To ensure balanced category representation, we grouped the MI-OAD validation image–caption pairs by category and manually selected 10,000 high-quality pairs (approximately 100 per category) to construct the MI-OAD test set. This careful manual filtering guarantees that MI-OAD is a dependable benchmark.

## 4  Experiments

In this section, we explore three key questions: 1) How can we effectively leverage the MI-OAD dataset? 2) How can we equip existing models with capabilities for open-set aerial detection? 3) How can we evaluate the open-set aerial detection capabilities of models at the word, phrase, and sentence levels? Additional experiments and implementation details are provided in the supplemental material.

## 4.1 MI-OAD Dataset Split and Sample

Base and Novel Classes Split. We designate 75 classes as Base and 25 classes as Novel. The class division is based on clustering the class semantic embeddings and selecting one class from each pair of leaf nodes in the clustering tree [30]. This assignment of novel classes ensures that the dataset can effectively evaluate zero-shot transfer capabilities.

Data Split. To fully exploit the available data while preserving the original splits of each detection dataset, we merge the train and test splits of all eight constituent datasets. Images containing only base categories form the pre-training set (P-Set), whereas the entire merged pool serves as the fine-tuning set (FT-Set). The validation splits are processed analogously: images that include at least one novel category (with only their novel annotations retained) constitute Val-ZSD, and the complete merged validation pool is denoted Val-FT. We use P-Set together with Val-ZSD to assess zero-shot transfer, while models fine-tuned on FT-Set are evaluated on Val-FT to benchmark conventional detection and grounding performance.

Sampling Strategy and Experimental Data Statistics. Considering the large scale of the dataset, the substantial computational resources required, and recognizing this as the first work focused on open-set aerial object detection, we conducted caption sampling post-annotation. Specifically, for each image, we categorized captions by type and then sampled one caption per type category to form image-caption pairs, ensuring dataset diversity and annotation quality. Consequently, the MI-OAD dataset comprises approximately 2 million image-caption pairs and 163,023 detection annotations. Specifically, The P-Set comprises 0.56M image–caption pairs and 68,243 detection annotations. The FT-Set include 1.40M pairs and 128 019 annotations. For validation: Val-ZSD provides about 0.12M pairs and 16,992 detection annotations for zero-shot evaluation, whereas Val-FT contains roughly 0.38M pairs and 35,004 annotations for conventional assessment.

## 4.2 Training Strategy

Most open-set detectors for natural images adopt the grounding data format introduced in [18]: each sample consists of an image–caption pair plus instance annotations, and a single image-level caption contains multiple noun phrases, each aligned with a distinct object. In aerial scenes, however, objects are densely packed and backgrounds are highly cluttered, making it infeasible to craft a caption that is both comprehensive and unambiguous for every instance.

To address this mismatch, we redefine the grounding format for aerial images. For each image we provide a set of instance-level captions; each caption describes one specific object (or a homogeneous group of objects) and is stored together with its bounding box. These fine-grained captions therefore extend the traditional notion of a category label with richer textual semantics.

Under this design we unify grounding and detection: the grounding task is recast as a detection task in which the instance-level caption replaces the corresponding class label. Consequently, the model learns open-set aerial detection while integrating linguistic cues, achieving a seamless combination of visual localization and textual classification.

## 4.3 Evaluation Details

To comprehensively evaluate open-set detection capability, we propose three evaluation protocols simulating real-world scenarios: vocabulary-level detection, phrase-level grounding, and sentence-level grounding, each corresponding to varying levels of detail in natural language input (vocabulary, phrase, and sentence). Additionally, we define three evaluation setups to assess detection performance under different constraints: zero-shot transfer to novel classes without domain adaptation, zero-shot transfer to novel classes with domain adaptation, and fine-tuned evaluation. The primary distinction between the first two setups is the use of the MI-OAD P-Set for domain adaptation of detectors originally designed for natural images.

## 4.4 Open-set Aerial Object Detection Results

From Table 1, we evaluate the open-set aerial detection capabilities of two representative approaches—Yolo-World (YOLOv8-L) and Grounding DINO (Swin-T)—across three different

| Method | Detection | | Phrase Grounding | | | | Sentence Grounding | | | |
|---|---|---|---|---|---|---|---|---|---|---|
| | $AP_{50}$ | R@100 | $AP_{50}$ | R@1 | R@10 | R@100 | $AP_{50}$ | R@1 | R@10 | R@100 |
| *Zero-shot transfer with novel classes (w/o domain adaptive).* | | | | | | | | | | |
| Yolo-World [3] | 3.2 | 37.1 | 3.8 | 6.8 | 25.0 | 34.4 | 1.4 | 4.3 | 16.9 | 24.6 |
| Grounding DINO [13] | 4.0 | 49.6 | 9.2 | 10.7 | 35.1 | 50.4 | 5.2 | 10.3 | 33.8 | 42.9 |
| *Zero-shot transfer with novel classes (w/ domain adaptive).* | | | | | | | | | | |
| Yolo-World | 5.3 | 30.6 | 18.0 | 18.3 | 43.5 | 55.9 | 15.9 | 19.1 | 44.9 | 57.1 |
| Grounding DINO | 9.8 | 69.8 | 32.1 | 24.1 | 60.9 | 80.9 | 36.3 | 35.1 | 68.5 | 82.7 |
| *Fine-tuned.* | | | | | | | | | | |
| Yolo-World | 39.6 | 58.0 | 51.6 | 32.9 | 69.9 | 86.4 | 47.6 | 36.1 | 71.1 | 86.9 |
| Grounding DINO | 37.1 | 70.1 | 57.8 | 35.2 | 74.4 | 91.5 | 56.4 | 44.1 | 78.0 | 90.3 |

Table 1: Performance comparison of representative methods on the MI-OAD dataset across different open-set evaluation tasks (vocabulary-level detection, phrase-level grounding, and sentence-level grounding). The evaluation setups differ as follows: zero-shot transfer w/ or w/o domain adaptation indicates whether the model was trained on the MI-OAD P-Set for domain adaptation, while fine-tuned conditions represent models trained on the FT-Set of MI-OAD.

evaluation scenarios. Both methods are evaluated on detection, phrase-level grounding, and sentence-level grounding, reflecting different levels of granularity in open-set detection tasks.

**Zero-shot Transfer (w/o Domain Adaptation).** When directly applying models trained on natural-image data to the MI-OAD V-Set, performance is notably limited. For instance, Yolo-World achieves a mere 1.4% $AP_{50}$ under sentence-level prompts. Grounding DINO performs slightly better (5.2% $AP_{50}$), yet both methods exhibit substantial performance gaps, demonstrating the unique challenges posed by open-set aerial object detection. **Zero-shot Transfer (w/ Domain Adaptation).** Introducing domain adaptation for these models by training on the MI-OAD P-Set results in considerable performance improvements for both methods. For example, Grounding DINO's detection $AP_{50}$ improve from 4.0% to 9.8%, while its sentence-level grounding $AP_{50}$ increases by 31.1%. These results underscore the effectiveness of our proposed dataset. **Fine-tuning.** After fine-tuning on the FT-Set, both models achieve superior results. Grounding DINO achieves outstanding performance, obtaining $AP_{50}$ values of 37.1% for detection, 57.8% for phrase grounding, and 56.4% for sentence grounding. These results demonstrate that the MI-OAD dataset provides an effective basis for advancing open-set aerial object detection and further confirm the importance of large-scale grounding data with rich textual annotations.

# 5 Conclusion

In this paper, we propose the OS-W2S Label Engine, which addresses the scarcity of rich textual grounding data in the aerial domain and establishes a robust data foundation for open-set aerial detection. Using this pipeline, we introduce the MI-OAD, the first benchmark dataset for open-set aerial detection. MI-OAD contains 163,023 images and 2.0 million image–caption pairs, with captions at the word, phrase, and sentence levels. We demonstrate that training existing open-set detectors on MI-OAD enables open-set aerial detection and improves performance across different caption levels. Our OS-W2S Label Engine and MI-OAD aim to benefit the research community and foster future advancements in open-set aerial detection.

# 6 Limitation and Broader impacts

The OS-W2S Label Engine and MI-OAD provide foundational resources to advance aerial object detection research, which can significantly benefit practical applications such as environmental monitoring and urban development planning. While these resources offer numerous advantages, we acknowledge two main limitations. (1) Even with rules to mitigate hallucinations from VLMs, the small sizes of aerial instances and occasional low-quality imagery can still lead to imprecise descriptions. (2) Our captions are constructed using only six fundamental attributes, constraining the range of details they can convey. By pointing out these limitations, we aim to stimulate future research towards generating richer and more precise captions for aerial imagery.

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
