# From Word to Sentence: A Large-Scale Multi-Instance Dataset for Open-Set Aerial Detection

## A    OS-W2S Label Engine Details

**Predefined Size and Absolute position Attributes.** Captions are generated based on six instance attributes: category, color, size, geometry, relative position, and absolute position. Among these attributes, the size (defined as the ratio of the instance's area to the image area) and absolute position (defined as the exact location of the instance within the image) are often subjectively determined. Moreover, the instances occupy only a very small portion of the image, which poses a challenge for the VLM to accurately determine the absolute positions of instances within the original images.

To address this issue, we apply predetermined rules to extract the size and absolute position attributes during the data pre-processing stage, explicitly providing this prior knowledge to the VLM during interaction. Specifically, we define size thresholds as [0.0005, 0.001, 0.01, 0.2], corresponding to bounding box area ratios relative to the image area, and categorize instances into ['tiny', 'small', 'medium', 'big', 'large']. Additionally, we segment the image into 25 regions using horizontal labels ['Far Left', 'Left', 'Center', 'Right', 'Far Right'] and vertical labels ['Top', 'Upper Middle', 'Middle', 'Lower Middle', 'Bottom'] to systematically define absolute position attributes.

**Foreground-extraction algorithm.** Aerial images typically contain numerous small, densely packed objects, making it difficult for a VLM to attend to the target instance when fed with the raw image. Benefiting from the precise bounding-box annotations available in our collected dataset, we are able to crop the corresponding foreground region for each instance, thereby effectively guiding the VLM's attention. However, to accurately characterise an instance's relative spatial attributes, it is also essential for the VLM to consider relationships between the instance, its surrounding context, and neighbouring objects. To address this, we design a foreground-extraction algorithm (Algorithm 1) that isolates salient regions while retaining sufficient context. This instance-level zoom-in strategy not only guides the VLM's attention more effectively but also leads to substantial improvements in caption quality.

**Matching Caption-Instance Pairs.** The captions generated by the VLM correspond specifically to single instances. Our objective is to clearly associate each caption with its corresponding instances based on the descriptive attributes included within each caption, as detailed in Algorithm 2. Specifically, since less detailed descriptions are more likely to match multiple instances, for each caption, we compare only the attributes explicitly mentioned in the caption with those of all instances present in the image. An instance is considered matched to a caption if the attributes share identical wording or their feature similarity exceeds a predetermined threshold; otherwise, no match is established.

## B    MI-OAD Dataset Details

**Base/Novel Categories split.** To ensure that the RSW2S dataset can effectively evaluate zero-shot transfer, we split the categories into 75 base categories and 25 novel categories. The class division is based on clustering the semantic embeddings of the classes and selecting one class from each pair of leaf nodes in the clustering tree [18]. The category splits are as follows:

Submitted to 39th Conference on Neural Information Processing Systems (NeurIPS 2025). Do not distribute.

**Algorithm 1** Foreground Region Extraction

---

**Require:** Bounding box set $B = \{b_1, b_2, \ldots, b_N\}$, image size $(w, h)$
**Ensure:** Foreground region set $R$
 1: **Step 1: Scale Bounding Boxes**
 2: **for** $i = 1$ to $N$ **do**
 3:     Compute the area $A_i$ of bounding box $b_i$
 4:     Determine scaling factor $s_i$ based on $A_i$
 5:     Update the bounding box $b_i$ to its extended version, ensuring it remains within the image boundaries
 6: **end for**
 7: **Step 2: Merge Overlapping Boxes**
 8: **for** each unmerged box $b_i$ **do**
 9:     Let $r \leftarrow b_i$
10:     **while** there exists an unmerged box $b_j$ that overlaps with $r$ **do**
11:         $r \leftarrow \text{MERGE}(r, b_j)$
12:         Mark $b_j$ as merged
13:     **end while**
14:     Add $r$ to the foreground region set $R$
15: **end for**
16: **return** $R$

---

**Algorithm 2** Caption-Instance Pair Matching

---

 1: **Input:**
 2: - `captions`: A list of captions, each containing textual descriptions and associated attributes (e.g., category, size, color, geometry, position).
 3: - `instances`: A list of object instances, each identified by an ID and associated attributes (e.g., category, size, color, geometry, position).
 4: **Output:**
 5: - `caption_instance_pairs`: A list of pairs $(caption, instance)$, where each caption is matched with corresponding object instances.
 6: **Step 1: Initialization**
 7: - Create an empty list `caption_instance_pairs` to store the matched caption-instance pairs.
 8: **Step 2: Matching Process**
 9: **for** each caption in `captions` **do**
10:     Extract relevant attributes (e.g., category, size, color) from the caption.
11:     Initialize an empty list `matched_instances` to store matching instances.
12:     **for** each instance in `instances` **do**
13:         Extract relevant attributes (e.g., category, size, color) from the instance.
14:         Compare attributes between caption and instance.
15:         **if** the attributes match sufficiently **then**
16:             Add `instance` to `matched_instances`.
17:         **end if**
18:     **end for**
19:     Add the pair $(caption, matched\_instances)$ to `caption_instance_pairs`.
20: **end for**
21: **Step 3: Output**
22: - Return `caption_instance_pairs`.

---

- **Base:** 'aircraft', 'aircraft-hangar', 'airplane', 'baseball-diamond', 'baseball-field', 'bicycle', 'bridge', 'building', 'car', 'cargo-car', 'cargo-plane', 'cargo-truck', 'cement-mixer', 'chimney', 'construction-site', 'container', 'container-crane', 'container-ship', 'crane-truck', 'dam', 'damaged-building', 'dump-truck', 'engineering-vehicle', 'expressway-service-area', 'expressway-toll-station', 'facility', 'ferry', 'fishing-vessel', 'fixed-wing-aircraft', 'flat-car', 'front-loader-or-bulldozer', 'golf-field', 'ground-grader', 'harbor', 'haul-truck', 'heli-pad', 'hut-or-tent', 'large-vehicle', 'locomotive', 'oil-tanker', 'overpass', 'passenger-car', 'passenger-vehicle', 'people', 'plane', 'pylon', 'railway-vehicle', 'roundabout', 'sailboat', 'shed', 'ship', 'shipping-container', 'small-aircraft', 'small-car', 'small-vehicle', 'soccer-ball-field', 'stadium', 'storage-tank', 'straddle-carrier', 'tank-car', 'tennis-court', 'tower', 'tower-crane', 'trailer', 'train-station', 'truck', 'truck-tractor', 'truck-tractor-with-flatbed-trailer', 'truck-tractor-with-liquid-tank', 'tugboat', 'utility-truck', 'van', 'vehicle', 'vehicle-lot', 'yacht'

- **Novel:** 'airport', 'awning-tricycle', 'barge', 'basketball-court', 'bus', 'crossroad', 'excavator', 'ground-track-field', 'helicopter', 'maritime-vessel', 'mobile-crane', 'motor', 'motor-boat', 'parking-lot', 'pedestrian', 'pickup-truck', 'playground', 'reach-stacker', 'scraper-or-tractor', 'shipping-container-lot', 'swimming-pool', 't-junction', 'tricycle', 'truck-tractor-with-box-trailer', 'windmill'

**MI-OAD Dataset Scale.** As shown in Table 1, we curated eight representative aerial detection datasets, and subsequently leveraged the OS-W2S Label Engine to enrich these datasets with comprehensive textual annotations, which collectively constitute the MI-OAD dataset. Specifically, MI-OAD comprises 163,023 images and 2,389,973 (2M) image–caption pairs. As summarized in Table 2, MI-OAD is approximately 40 times larger than existing remote sensing grounding datasets and offers substantially higher annotation quality.

Table 1: Overview of the collected aerial-detection datasets. Image and instance counts are reported after cropping to a uniform resolution.

| Dataset | Images | Instances | Categories |
|---|---|---|---|
| DIOR [5] | 23,463 | 192,518 | 20 |
| DOTA v2.0 [12] | 19,871 | 495,754 | 18 |
| HRRSD [20] | 44,002 | 96,387 | 13 |
| NWPU_VHR_10 [10] | 1,244 | 6,778 | 10 |
| RSOD [13] | 3,644 | 22,221 | 4 |
| SODA-A [8] | 31,798 | 1,008,346 | 9 |
| VisDrone [21] | 29,040 | 740,419 | 10 |
| xView [4] | 9,961 | 732,960 | 60 |

Table 2: Comparison with existing remote-sensing grounding datasets.

| Dataset | Categories | Images | Image–Caption Pairs |
|---|---|---|---|
| RSVG-H [11] | – | 4,239 | 7,933 |
| DIOR-RSVG [19] | 20 | 17,402 | 38,320 |
| OPT-RSVG [6] | 14 | 25,452 | 48,952 |
| MI-OAD (ours) | 100 | 163,023 | 2M |

# C  More Experimental Results

In this section, we conduct additional experiments to comprehensively demonstrate the advantages of the proposed MI-OAD dataset. We show that the MI-OAD dataset is not only suitable for open-set aerial object detection but also provides substantial benefits for open-vocabulary detection and remote-sensing visual grounding tasks.

## C.1 Training Details.

### C.1.1 Baselines and experimental setup.

To demonstrate the effectiveness of MI-OAD, we conduct experiments on two representative tasks: (i) open-set aerial object detection and (ii) remote-sensing visual grounding (RSVG).

For open-set aerial object detection, we evaluate two representative open-set detectors—Grounding DINO [7] and YOLO-World [1]—on MI-OAD at three semantic granularities: *vocabulary*, *phrase*, and *sentence*. This constitutes the first comprehensive benchmark for open-set aerial object detection. We adopt the MMDetection implementation of Grounding DINO and the official v1.0 release of YOLO-World. Unless otherwise specified, all experiments are executed on 32 NVIDIA RTX 4090 GPUs with a batch size of four per GPU. Grounding DINO is trained for 12 epochs, whereas YOLO-World is trained for 40 epochs; all other hyper-parameters remain at their default values.

For remote-sensing visual grounding (RSVG), we use Grounding DINO as the baseline and evaluate it on two standard benchmarks: DIOR-RSVG and OPT-RSVG. We first report performance without MI-OAD pre-training, and subsequently examine the model pre-trained on MI-OAD and fine-tuned separately on each benchmark. All experiments are conducted on eight NVIDIA RTX 4090 GPUs for 12 epochs with a batch size of four per GPU, while keeping all other hyper-parameters at their default settings. During evaluation, we retain only the bounding box with the highest confidence score and report standard metrics (Pr@{0.5, 0.6, 0.7, 0.8, 0.9}, mean IoU, and cmu IoU). We also compare the resulting scores with those of current state-of-the-art methods, including MGVLF [19] and LPVA [6], to quantify the gains afforded by MI-OAD pre-training.

### C.1.2 Prompt Construction Strategy

Prompt construction plays a crucial role in both training and inference phases. To enhance model robustness, we apply a randomized category sampling strategy during training. Specifically, for each detection sample, we define categories present in the image as positive classes ($C_{pos}$) and consider the remaining categories as negative ($C_{neg}$). We include all positive classes and randomly select between 1 and $|C_{neg}|$ negative classes to form the textual prompt associated with each sample.

However, since MI-OAD integrates eight distinct detection datasets, category conflicts across datasets may arise. For instance, an image containing an object labeled as *airplane* should consider *airplane* as a positive class; however, related categories such as *aircraft* could incorrectly appear among negative classes. To prevent such conflicts, we restrict negative class sampling strictly to categories from the same original dataset, as these annotations are manually verified and thus consistent. For grounding samples, we adopt a consistent approach by simply replacing the positive class labels with the corresponding image captions.

During inference, detection samples utilize prompts consisting of all categories from their respective source datasets, while grounding samples use prompts composed solely of their corresponding captions. For the RSVG tasks, prompts used during both training and inference exclusively consist of the captions corresponding to their respective images, independent of the previously described category sampling strategy.

## C.2 Performance on Open-Set Aerial Detection

As detailed in Section 4, experiments on the MI-OAD validation set demonstrate that the proposed dataset substantially enhances performance in open-set aerial object detection. To establish a rigorous benchmark for this task, we further evaluate Grounding DINO and YOLO-World on the manually verified MI-OAD test set. The results are summarised in Table 3. Because the detection annotations have already been manually verified, the MI-OAD test set focuses exclusively on grounding tasks with phrase- and sentence-level inputs.

## C.3 Performance on Remote-Sensing Visual Grounding

We further to explore the potential of MI-OAD for remote-sensing visual grounding (RSVG). Table 4 and Table 5 report Grounding DINO's performance on the OPT-RSVG and DIOR-RSVG test sets under two training paradigms: (i) training on each benchmark only, (ii) MI-OAD pre-training followed by task-specific fine-tuning.

Table 3: Performance on the MI-OAD test set across two open-set tasks: phrase-level grounding and sentence-level grounding.

| Method | Phrase Grounding | | | | Sentence Grounding | | | |
|--------|-----------------|--|--|--|-------------------|--|--|--|
| | $AP_{50}$ | R@1 | R@10 | R@100 | $AP_{50}$ | R@1 | R@10 | R@100 |
| *Zero-shot transfer, novel classes (with domain adaptation)* | | | | | | | | |
| YOLO-World | 19.5 | 18.6 | 42.3 | 55.0 | 16.4 | 19.7 | 43.7 | 55.4 |
| Grounding DINO | 33.2 | 24.4 | 60.3 | 81.1 | 37.6 | 35.4 | 68.8 | 82.7 |
| *Fine-tuned* | | | | | | | | |
| YOLO-World | 52.7 | 34.2 | 70.8 | 87.8 | 47.9 | 36.0 | 71.4 | 86.6 |
| Grounding DINO | 58.3 | 35.7 | 75.3 | 92.2 | 57.3 | 44.0 | 78.0 | 89.7 |

Table 4: Comparison with state-of-the-art methods on the OPT-RSVG test set (English version).

| Method | Pr@0.5 | Pr@0.6 | Pr@0.7 | Pr@0.8 | Pr@0.9 | meanIoU | cmuIoU |
|--------|--------|--------|--------|--------|--------|---------|--------|
| *One-stage* | | | | | | | |
| ZSGNet (ICCV'19) [9] | 48.64 | 47.32 | 43.85 | 27.69 | 6.33 | 43.01 | 47.71 |
| FAOA (ICCV'19) [15] | 68.13 | 64.30 | 57.15 | 41.83 | 15.33 | 58.79 | 65.20 |
| ReSC (ECCV'20) [16] | 69.12 | 64.63 | 58.20 | 43.01 | 14.85 | 60.18 | 65.84 |
| LBYL-Net (CVPR'21) [3] | 70.22 | 65.39 | 58.65 | 37.54 | 9.46 | 60.57 | 70.28 |
| *Transformer-based* | | | | | | | |
| TransVG (CVPR'21) [2] | 69.96 | 64.17 | 54.68 | 38.01 | 12.75 | 59.80 | 69.31 |
| QRNet (CVPR'22) [17] | 72.03 | 65.94 | 56.90 | 40.70 | 13.35 | 60.82 | 75.39 |
| VLTGV (ResNet-50) (CVPR'22) [14] | 71.84 | 66.54 | 57.79 | 41.63 | 14.62 | 60.78 | 70.69 |
| VLTGV (ResNet-101) (CVPR'22) [14] | 73.50 | 68.13 | 59.93 | 43.45 | 15.31 | 62.48 | 73.86 |
| MGVLF (TGRS'23) [19] | 72.19 | 66.86 | 58.02 | 42.51 | 15.30 | 61.51 | 71.80 |
| LPVA (TGRS'24) [6] | 78.03 | 73.32 | 62.22 | 49.60 | 25.61 | 66.20 | 76.30 |
| Grounding DINO (OPT-RSVG Train) | 75.73 | 72.62 | 66.30 | 53.29 | 28.63 | 65.66 | 71.12 |
| *Pretrained on MI-OAD, fine-tuned on OPT-RSVG* | | | | | | | |
| Grounding DINO | **82.62** | **80.83** | **76.59** | **65.26** | **38.13** | **72.61** | **77.00** |
| Gain over GD (OPT-RSVG Train) | +6.9 | +8.2 | +10.3 | +11.9 | +9.5 | +6.9 | +5.9 |
| Gain over LPVA (SOTA) | +4.6 | +7.5 | +14.4 | +15.7 | +12.5 | +6.4 | +0.7 |

Pre-training on MI-OAD boosts Grounding DINO's Pr@0.9 from 28.6 % to 38.1 % and mean IoU from 65.7 % to 72.6 % on OPT-RSVG, surpassing the previous best LPVA by up to 15.7 % (Pr@0.8) and 6.4 % in mean IoU, and establishing state-of-the-art performance on OPT-RSVG. On DIOR-RSVG, the same pre-training raises Pr@0.9 from 44.2 % to 49.3 % and mean IoU from 70.0 % to 74.5 %, outperforming LPVA by 9.7 % and 2.2 %, respectively.

These results confirm that MI-OAD significantly strengthens the model's grounding ability, particularly at high-IoU thresholds where precise localisation is critical. The larger pre-training corpus consistently yields higher precision and mean IoU scores, revealing a clear scaling-law trend: as the amount of pre-training data increases, the model's robustness and generalization improve. This observation underscores the importance of both our OS-W2S Label Engine and the diverse MI-OAD dataset for advancing remote-sensing visual grounding.

## D   Qualitative Analysis of Open-Set Aerial Detection Results

In this section, we demonstrate and analyze the effectiveness of our proposed dataset from three perspectives. First, we compare and visualize the detection results of Grounding DINO with and without domain adaptation using our MI-OAD P-Set dataset. Second, to simulate realistic application scenarios, we evaluate the model's open-set aerial detection capability trained on the MI-OAD dataset using self-defined prompts that do not originate from annotation files. Finally, we visualize the

Table 5: Comparison with state-of-the-art methods on the DIOR-RSVG test set (English version).

| Method | Pr@0.5 | Pr@0.6 | Pr@0.7 | Pr@0.8 | Pr@0.9 | meanIoU | cmuIoU |
|---|---|---|---|---|---|---|---|
| *One-stage* | | | | | | | |
| ZSGNet (ICCV'19) [9] | 51.67 | 48.13 | 42.30 | 32.41 | 10.15 | 44.12 | 51.65 |
| FAOA (ICCV'19) [15] | 67.21 | 64.18 | 59.23 | 50.87 | 34.44 | 59.76 | 63.14 |
| ReSC (ECCV'20) [16] | 72.71 | 68.92 | 63.01 | 53.70 | 33.37 | 64.24 | 68.10 |
| LBYL-Net (CVPR'21) [3] | 73.78 | 69.22 | 65.56 | 47.89 | 15.69 | 65.92 | 76.37 |
| *Transformer-based* | | | | | | | |
| TransVG (CVPR'21) [2] | 72.41 | 67.38 | 60.05 | 49.10 | 27.84 | 63.56 | 76.27 |
| QRNet (CVPR'22) [17] | 75.84 | 70.82 | 62.27 | 49.63 | 25.69 | 66.80 | 83.02 |
| VLTGV (R-50) (CVPR'22) [14] | 69.41 | 65.16 | 58.44 | 46.56 | 24.37 | 59.96 | 71.97 |
| VLTGV (R-101) (CVPR'22) [14] | 75.79 | 72.22 | 66.33 | 55.17 | 33.11 | 66.32 | 77.85 |
| MGVLF (TGRS'23) [19] | 75.98 | 72.06 | 65.23 | 54.89 | 35.65 | 67.48 | 78.63 |
| LPVA (TGRS'24) [6] | 82.27 | 77.44 | 72.25 | 60.98 | 39.55 | 72.35 | **85.11** |
| Grounding DINO (DIOR-RSVG Train) | 77.85 | 75.69 | 71.14 | 62.65 | 44.19 | 69.96 | 79.36 |
| *Pretrained on MI-OAD, fine-tuned on DIOR-RSVG* | | | | | | | |
| Grounding DINO | **82.46** | **80.92** | **77.43** | **69.20** | **49.26** | **74.51** | 81.69 |
| Gain over GD (DIOR-RSVG Train) | +4.61 | +5.23 | +6.29 | +6.55 | +5.07 | +4.55 | +2.33 |
| Gain over LPVA (SOTA) | +0.19 | +3.48 | +5.18 | +8.22 | +9.71 | +2.16 | -3.42 |

model's performance on the MI-OAD V-Set by employing prompts at three input levels: vocabulary-level, phrase-level, and sentence-level.

## D.1 Comparison of Grounding DINO with and without Domain Adaptation

Fig. 1 visualizes the detection results of Grounding DINO before and after domain adaptation training on our MI-OAD P-Set dataset. As observed in the results of the first and second columns, Grounding DINO, originally designed for natural images, yields suboptimal performance when directly applied to aerial imagery domains. However, after domain adaptation using our proposed dataset, the detection results significantly improve. From the third column, we observe that while Grounding DINO can localize objects in common urban scenarios, it exhibits clear false positives and misses detections—for example, incorrectly detecting a green taxi with the prompt "white car" and missing smaller white cars in the distance. Following training on our dataset, the model notably improves its ability to detect smaller instances and accurately recognize instance attributes.

## D.2 Evaluation of Open-Set Aerial Detection Using Self-Defined Prompts

To further demonstrate the practical efficacy of our dataset, we simulate realistic scenarios by employing self-defined prompts (rather than those derived from annotated files) to evaluate the open-set aerial detection capability of Grounding DINO after domain adaptation using our proposed datasets.

Fig. 2 visualizes detection results from the model in urban and harbor scenarios. Notably, in the second column of the first row, the model successfully detects objects described by implicitly defined prompts, where object categories are not explicitly mentioned but described solely through attributes. This capability can be attributed to the attribute-based captions in our dataset and Grounding DINO's sentence-level image-text alignment approach. Additional examples also highlight the model's sensitivity to relative and absolute positional information.

As illustrated in Fig. 3, we further evaluate the model's open-set aerial detection performance using different levels of prompts, ranging from single words to phrases, and ultimately sentences. To intuitively illustrate the influence of different prompt complexities on model performance, we conducted tests on the same image. The first column in the first row of Fig.3 demonstrates the model's strong generalization capability, accurately detecting objects corresponding to prompts including novel classes, such as "a green taxi on the street." Moreover, the model shows strong sensitivity to relative positional attributes, as highlighted in the third column of the first row, where two white

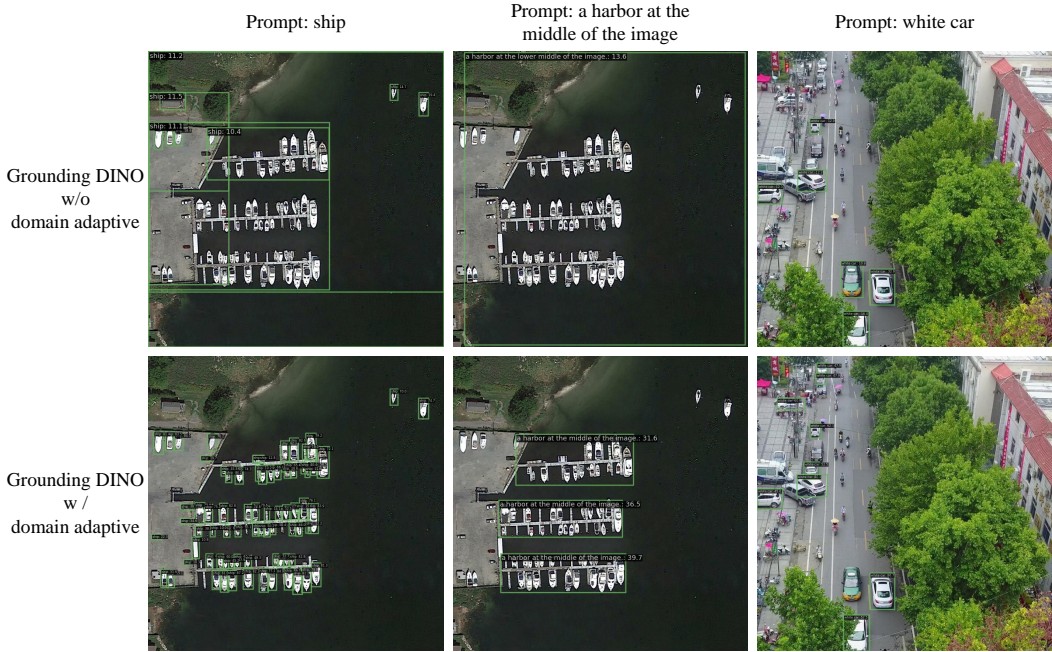

Figure 1: Visualization of detection results comparing GroundingDINO without and with domain adaptation using our proposed MI-OAD P-Set dataset.

cars near a green taxi are accurately identified based on the prompt "right of the green taxi," further indicating the model's spatial understanding capability. Additionally, as shown in the third row, the model achieves highly precise detection results for small targets.

## D.3 Visualization of Detection Results at Vocabulary, Phrase, and Sentence Levels

We also visualize the model's open-set detection capability on the MI-OAD V-Set using prompts derived from annotations. Specifically, we illustrate the detection performance at three different prompt complexity levels: vocabulary-level, as shown in Fig.4; phrase-level, as shown in Fig.5; and sentence-level, as shown in Fig. 6.

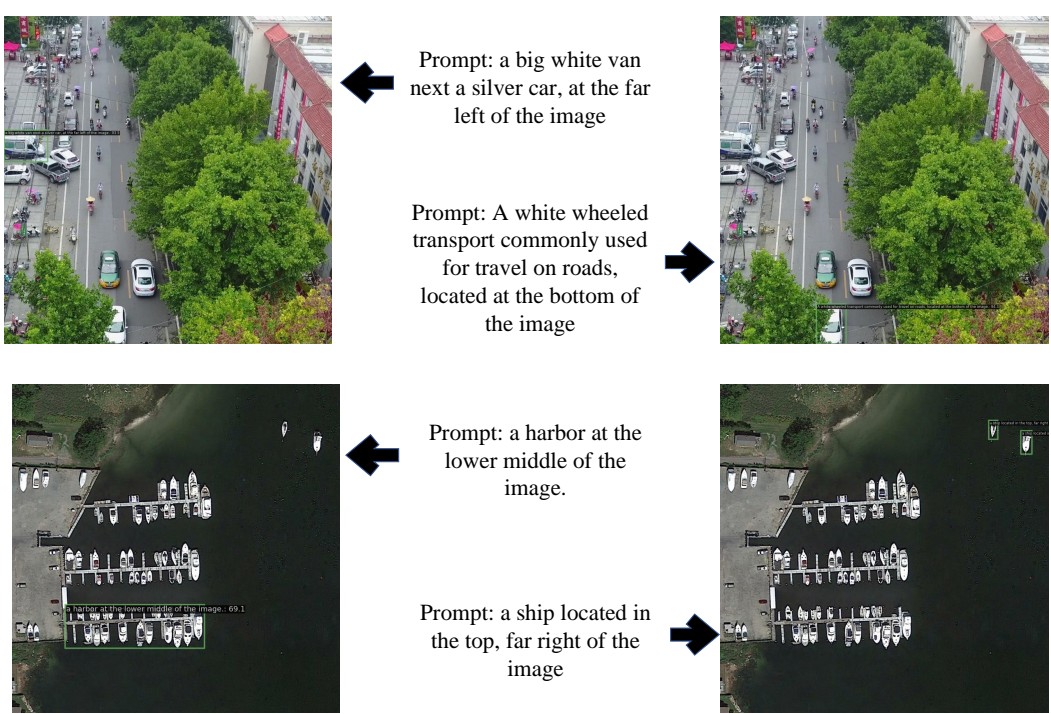

Figure 2: Qualitative visualization of open-set aerial detection performance with self-defined prompts (Part 1)

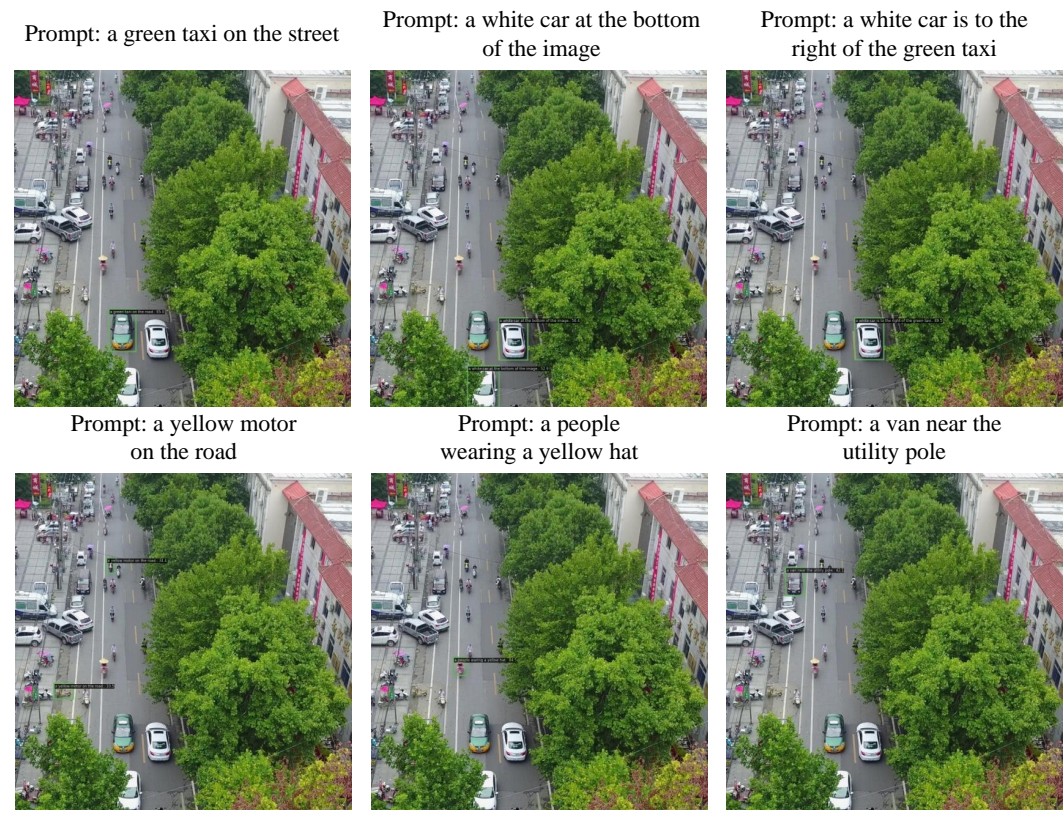

Figure 3: Qualitative visualization of open-set aerial detection performance with self-defined prompts (Part 2)

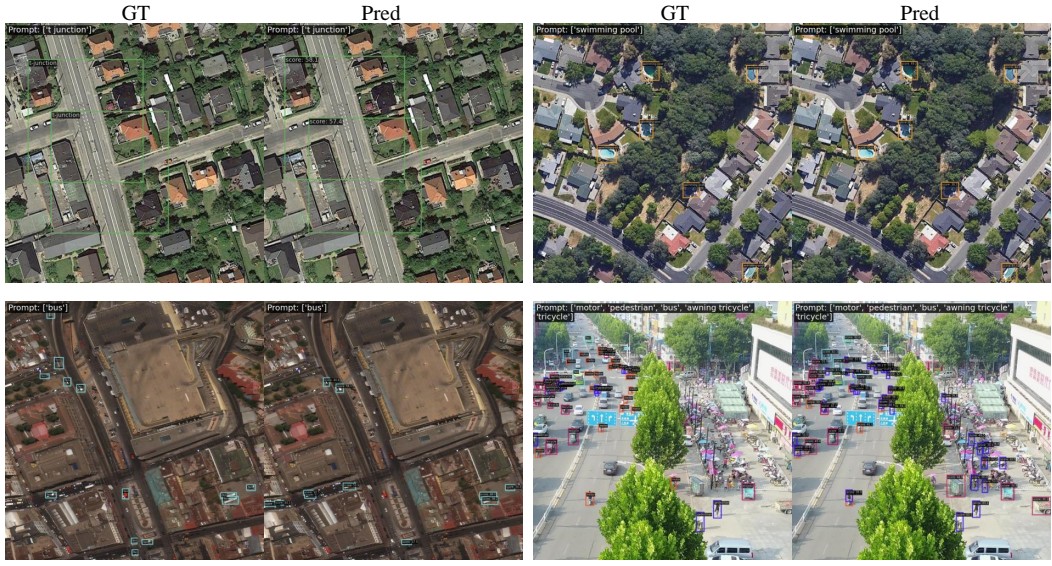

Figure 4: Visualization of open-set aerial detection results at the vocabulary-level using annotation-derived prompts.

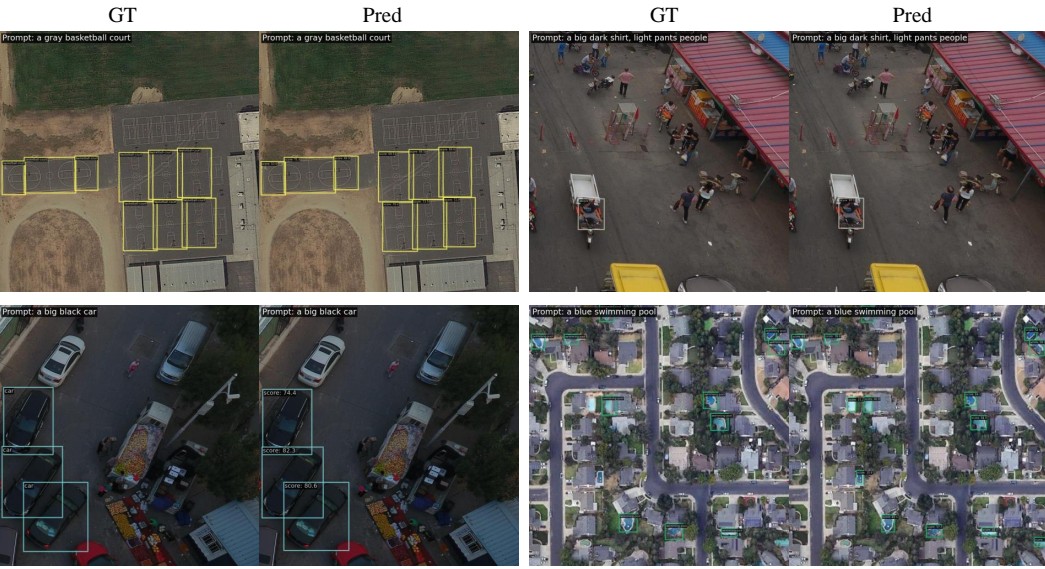

Figure 5: Visualization of open-set aerial detection results at the phrase-level using annotation-derived prompts.

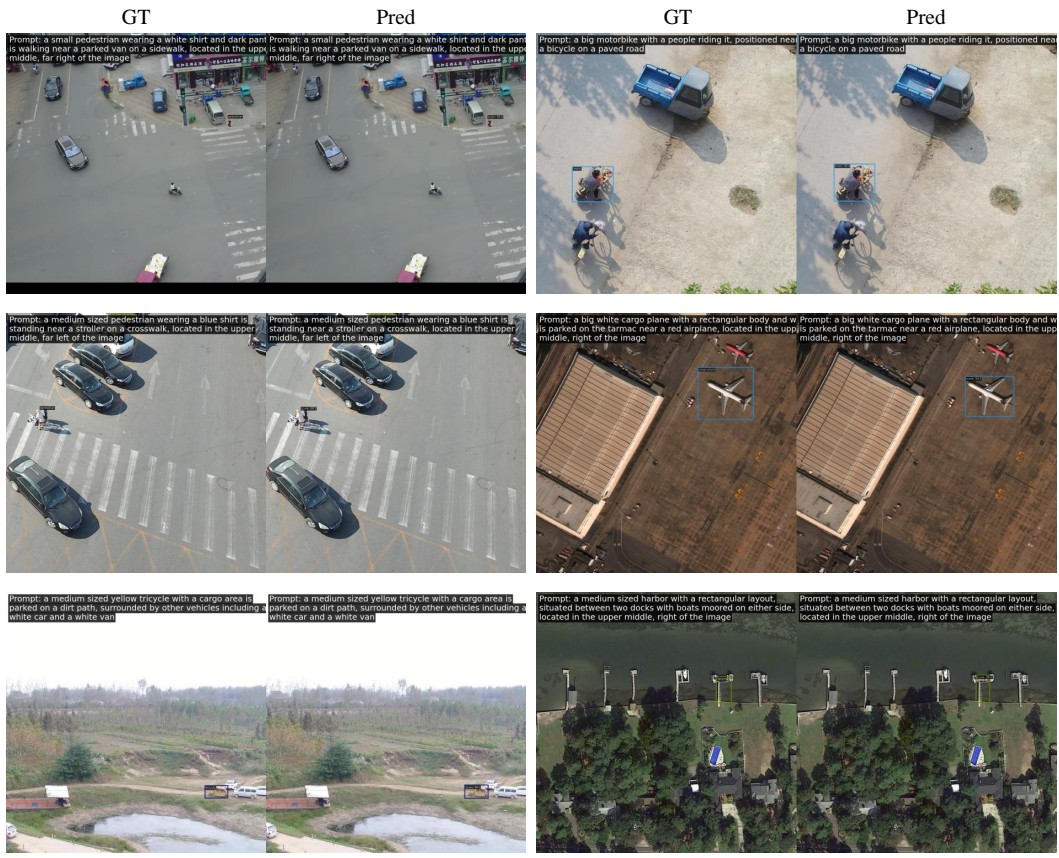

Figure 6: Visualization of open-set aerial detection results at the sentence-level using annotation-derived prompts.

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

**Important**: The red box in the provied images is only for your reference to identify the target. **Do not mention the red box or any red box-related information in final caption.**

Now, let's start with **Step 1**:

Prompt2:

```
caption1_template = {
    "caption": f"[A brief sentence describing the target using the provided Category and Size.
Include **Color** and **Geometry** only if you are certain about them.]",
    "Category": f"{gt_category}",
    "Size": f"{gt_size}",
    "Color": "[Include if certain]",
    "Geometry": "[Include if certain]"
}
```
<image>

**Step 1**: You are provided with an aerial image of a target. The red box highlights the target.

- Generate a caption describing the target.
- **Must** using the provided **Category:** "{gt_category}" and **Size:** "{gt_size}" in caption.
- Include **Color** and **Geometry** only if you are certain about them.
- Do **not** mention the red box or any red box-related information in final caption.
- Keep the caption under 20 words.
- Only include information you can confidently determine from the image. Avoid speculative or aesthetic descriptions.

**Must format your answer as a JSON object with the following structure and strictly adhere to the JSON format:**

{caption1_template}

Prompt3:
caption2_template = {
    "caption": f"[Refined caption including the target's attributes and relative location.]",
    "relative_location": "[The target's relative location within its surroundings.]"
}
<image>
**Step 2**: You are provided with a cropped section of an aerial image showing the target within its surroundings. The red box highlights the target (for your reference, do not mention it).

- Based on the caption from Step 1: "{self_caption}", refine the description by incorporating **relative location** information about the target with respect to its surrounding environment or nearby objects.
- Maintain the original attributes (**Category**, **Size**, **Color**, **Geometry**).
- Do **not** mention the red box or any red box-related information in final caption.
- Do **not** describe the target's location relative to the image position (e.g., 'top left of the image').
- Keep the caption under 40 words.
- Only include information you can confidently determine from the image. Avoid speculative or aesthetic descriptions.

**Must format your answer as a JSON object with the following structure and strictly adhere to the JSON format:**
{caption2_template}

Prompt4:
caption3_template = {
    "caption": "[The caption by incorporating the absolute location.]",
    "absolute_location": f"{box_pos}"
}
**Step 3**: You are provided the region of the image where the target is located.

- Review the caption from Step 2: "{relative_caption}", enhance the caption by incorporating the provided **absolute location** information.
**absolute Location**: "{box_pos}"

- Keep the caption under 60 words.
- Only include information you can confidently determine from the image. Avoid speculative or aesthetic descriptions.

**You must format your answer as a JSON object with the following structure and strictly adhere

to the JSON format:**
{caption3_template}