# OpenReview forum: "From Word to Sentence: A Large-Scale Multi-Instance Dataset for Open-Set Aerial Detection"
_NeurIPS.cc/2025/Datasets_and_Benchmarks_Track — Submitted to NeurIPS 2025 Datasets and Benchmarks Track_

### Official Review · Reviewer_E4d1 · 2025-06-27

**Rating:** 4
**Confidence:** 3

**Summary:**

This paper introduces MI-OAD , the first large-scale benchmark dataset for open-set aerial object detection, addressing critical limitations in existing remote sensing visual grounding (RSVG) datasets. The authors propose the OS-W2S Label Engine , an automated annotation pipeline leveraging VLMs to generate multi-instance annotations. MI-OAD contains 163,023 images and 2 million image-caption pairs, significantly surpassing existing datasets in scale and diversity. The dataset is validated using state-of-the-art open-set detection models achieving substantial performance gains under zero-shot and fine-tuned settings.

**Dataset Code Accessibility:**

Yes

**Dataset Code Comments:**

The authors provided the dataset and code on Kaggle and GitHub, respectively.

**Ethical Considerations:**

No, there are no or only very minor ethics concerns

**Final Justification:**

After considering all reviews, responses, rebuttals, and discussions, I have decided to maintain my score.

**Limitations Weaknesses:**

-  MI-OAD aggregates eight aerial detection datasets. If these sources have inherent biases (e.g., overrepresentation of certain object types), MI-OAD may inherit them.
- Only two models (YOLO-World and Grounding DINO) are evaluated. Testing additional frameworks would better generalize conclusions.
- Using only six attributes (category, size, color, geometry, relative/absolute position) limits expressive richness. Incorporating texture or contextual relationships could enhance caption diversity.

**Strengths Contributions:**

- Existing RSVG datasets suffer from limited scene diversity, template-based captions, single-instance annotations, and small scale. MI-OAD resolves these issues with a 40x larger dataset, multi-instance support, and rich semantic descriptions, enabling practical open-set aerial detection.
- Training Grounding DINO on MI-OAD achieves a 31.1 AP50 improvement under zero-shot transfer, validating the dataset’s utility.
- The focus on open-set detection aligns with real-world scenarios.

---

> ### Author Rebuttal · Authors · 2025-07-31
>
> We greatly appreciate your detailed feedback and insightful comments. To address your concerns, we provide point-to-point responses below:
>
> > Comment 1: MI-OAD aggregates eight aerial detection datasets. If these sources have inherent biases (e.g., overrepresentation of certain object types), MI-OAD may inherit them.
>
> We fully understand your concern regarding the potential for source-specific biases to be inherited by MI-OAD. Our approach to mitigate these concerns includes the following measures:
>
> - **Dataset Selection:**  We carefully selected eight widely-used aerial detection datasets, all characterized by rigorous human annotation processes and high annotation quality. These datasets have become standard benchmarks within the aerial detection community, significantly reducing the risk of potential biases.
>
> - **Importance of Comprehensive Data for Open-Set Detection:** Open-set aerial detection is fundamentally data-driven, requiring large-scale and semantically rich datasets to achieve effective generalization. Given the severe scarcity of diverse grounding data in this field, each data instance is critically important. Thus, we deliberately refrained from further filtering beyond our initial rigorous dataset selection, preserving the semantic and visual diversity.
> - **Preservation of Source Metadata:** Throughout the construction of MI-OAD, we have preserved comprehensive metadata from the original source datasets, including dataset names and image IDs. Researchers can use this metadata to trace and selectively filter or customize the dataset according to their specific requirements.
>
> ---
>
> > Comment 2: Only two models (YOLO-World and Grounding DINO) are evaluated. Testing additional frameworks would better generalize conclusions.
>
> We appreciate your suggestion. Below we sequentially clarify why our initial evaluation included only YOLO-World and Grounding DINO and present additional experiments to strengthen our conclusions.
>
> - **Reasons for Initially Evaluating Only Two Models**
>
> Initially, we employed a rigorous Base/Novel class split to clearly assess zero-shot transfer capabilities. Existing open-vocabulary aerial object detection models, such as LAE-DINO [1], do not adopt this strict Base/Novel class division. Evaluating these models directly on our benchmark would result in class-overlap and information leakage (e.g., LAE-1M used by LAE-DINO contains categories that overlap with MI-OAD’s Novel classes). This overlap will inflates metrics, as illustrated in Table 1, especially in Detection and Phrase Grounding tasks. Consequently, directly incorporating methods in our benchmark comparison would be unfair to other methods.
>
> **Table 1. Performance comparison of representative methods on the MI-OAD dataset (Zero-shot transfer with novel classes, without domain adaptation).**
>
> | Method             | Detection AP₅₀ | Detection R@100 | Phrase AP₅₀ | Phrase R@1 | Phrase R@10 | Phrase R@100 | Sentence AP₅₀ | Sentence R@1 | Sentence R@10 | Sentence R@100 |
> | ------------------ | -------------- | --------------- | ----------- | ---------- | ----------- | ------------ | ------------- | ------------ | ------------- | -------------- |
> | Yolo-World [2]     | 3.2            | 37.1            | 3.8         | 6.8        | 25.0        | 34.4         | 1.4           | 4.3          | 16.9          | 24.6           |
> | Grounding DINO [3] | 4.0            | 49.6            | 9.2         | 10.7       | 35.1        | 50.4         | 5.2           | 10.3         | 33.8          | 42.9           |
> | LAE-DINO [1]       | 28.7           | 68.5            | 24.4        | 22.5       | 53.4        | 76.2         | 7.8           | 18.5         | 46.5          | 67.0           |
>
> - **Expanded Experimental Results**
>
> We conducted additional experiments incorporating the LAE-DINO model to strengthen our conclusions. Results in Table 2 demonstrate that incorporating MI-OAD during pre-training notably enhances performance across several open-vocabulary aerial detection benchmarks.
>
> **Table 2. Open-Vocabulary aerial detection performance on DIOR, DOTAv2.0, and LAE-80C benchmarks.**
>
> | Training Status | Method   | Pre-Training Data | DIOR AP₅₀ | DOTAv2.0 mAP | LAE-80C mAP |
> | --------------- | -------- | ----------------- | --------- | ------------ | ----------- |
> | Fully Trained   | LAE-DINO | LAE-1M            | 85.5      | 46.8         | 20.2        |
> | Training        | LAE-DINO | LAE-1M            | 83.4      | 45.3         | 17.8        |
> | Training        | LAE-DINO | + MI-OAD          | 91.6      | 50.3         | 20.5        |
>
> **Note:** “Fully Trained” results reflect the original reported metrics, where the model was thoroughly trained for 32 epochs with 4×A100 GPUs. Due to time constraints, the “Training” results are from models trained for 13 epochs with 32 RTX 4090 GPUs; thus, these models were not fully converged. Nevertheless, after incorporating the MI-OAD dataset, the performance already surpasses the “Fully Trained” baseline, further demonstrating the effectiveness of our dataset.
>
> **Action:** Following your suggestions, the LAE-1M experiment will be separately detailed in supplementary materials due to space constraints in the main text.
>
> ---
>
> > Comment 3: Using only six attributes (category, size, color, geometry, relative/absolute position) limits expressive richness. Incorporating texture or contextual relationships could enhance caption diversity.
>
> We appreciate your concern regarding the potential limitations in expressive richness when using only six attributes. We would like to clarify our rationale for selecting these specific attributes:
>
> - **Reference to Prior Datasets:** Previous remote sensing visual grounding datasets, such as DIOR-RSVG [4] and OPT-RSVG [5], included seven attributes: category, size, color, geometry, relative/absolute position, and relative size relation. These datasets were constructed under restrictive conditions (e.g., limiting images to no more than five objects per category), resulting in relatively simple scenes where attributes like “relative size relation” were practical and effective. In contrast, MI-OAD retains all images, including complex scenarios with dense and similarly sized instances. Consequently, attributes like relative size relation become challenging to annotate reliably.
> - **Insights from a Recent Study:** A recent study, ATPrompt [6], introduced an effective differentiable attribute search method to identify appropriate attributes tailored for specific datasets. The study demonstrated that even fewer attributes could effectively describe complex datasets (e.g., ImageNet: color, shape; Caltech101: shape, size; Flowers102: color, habitat, growth), indirectly supporting our approach that six independent attributes are sufficient to cover essential semantic diversity.
> - **Risk of Redundancy and Hallucinations:** Introducing additional attributes may increase redundancy and the risk of hallucinations by LVLMs. For example, “texture” might overlap with “geometry,” as LVLMs often embed texture information within geometry descriptions. Similarly, “contextual relationships” can overlap with “relative position.”
>
> Therefore, to balance comprehensiveness and avoid redundancy or conflicts, we ultimately selected six robust attributes (category, size, color, geometry, relative/absolute position) that can be reliably annotated across diverse scenes. Nevertheless, exploring additional attributes to further enrich expressive diversity is a promising direction for future work, potentially achievable through carefully designed attribute-selection methods.
>
> ---
>
> **References:**
>
> [1] Pan J, Liu Y, Fu Y, et al. Locate anything on earth: Advancing open-vocabulary object detection for remote sensing community[C]//Proceedings of the AAAI Conference on Artificial Intelligence. 2025, 39(6): 6281-6289.
>
> [2] Cheng T, Song L, Ge Y, et al. Yolo-world: Real-time open-vocabulary object detection[C]//Proceedings of the IEEE/CVF conference on computer vision and pattern recognition. 2024: 16901-16911.
>
> [3] Liu S, Zeng Z, Ren T, et al. Grounding dino: Marrying dino with grounded pre-training for open-set object detection[C]//European conference on computer vision. 2024: 38-55.
>
> [4] Zhan Y, Xiong Z, Yuan Y. Rsvg: Exploring data and models for visual grounding on remote sensing data[J]. IEEE Transactions on Geoscience and Remote Sensing, 2023, 61: 1-13.
>
> [5] Li K, Wang D, Xu H, et al. Language-guided progressive attention for visual grounding in remote sensing images[J]. IEEE Transactions on Geoscience and Remote Sensing, 2024, 62: 1-13.
>
> [6] Li Z, Song Y, Zhao P, et al. ATPrompt: Textual Prompt Learning with Embedded Attributes[J]. arXiv preprint arXiv:2412.09442, 2024.
>
> ---

---

> > ### Comment · Reviewer_E4d1 · 2025-08-05
> >
> > Thank you for the authors’ response. I remain concerned about the contribution of this work, as it primarily aggregates eight aerial detection datasets and the evaluation methods seem limited.

---

> > > ### Author Response · Authors · 2025-08-05
> > >
> > > Thank you very much for your valuable feedback, which has been instrumental in further improving our work. Below, we clarify your concerns comprehensively from the perspectives of contributions, dataset construction, and evaluation methodology.
> > >
> > > ### Contributions of Our Work:
> > >
> > > We would like to clarify that our primary contribution is not merely aggregating eight existing aerial detection datasets. Rather, it is the development of the OS-W2S Label Engine, an innovative pipeline leveraging LVLMs to construct a robust data foundation for open-set aerial object detection.
> > >
> > > - **OS-W2S Label Engine:** We propose an automated annotation pipeline that effectively leverages LVLMs to significantly enrich semantically sparse aerial detection datasets. Our pipeline provides multi-level language guidance—from words to phrases, and ultimately to sentences—and can be replicated using commonly accessible hardware (e.g., a workstation with eight RTX 4090 GPUs), ensuring practicality and reproducibility.
> > > - **Data Foundation for Open-Set Aerial Detection:** Through the OS-W2S Label Engine, we expand upon eight rigorously annotated and widely used aerial detection datasets, resulting in MI-OAD—the first open-set aerial object detection benchmark. It comprises 163,023 images and 2 million image-caption pairs, making it approximately 40 times larger than existing comparable datasets. This substantial dataset provides a foundational resource for advancing open-set aerial detection research.
> > >
> > > ### Considerations for Dataset Acquisition:
> > >
> > > **To ensure benchmark quality**, we deliberately selected eight widely recognized aerial detection datasets known for their rigorous annotation processes and high-quality human annotations. We acknowledge that limiting dataset selection might slightly constrain the dataset's scale; however, **our OS-W2S Label Engine is easily reproducible, enabling researchers to seamlessly incorporate additional annotated datasets in the future.**
> > >
> > > ### Evaluation of MI-OAD across Multiple Tasks:
> > >
> > > We demonstrated MI-OAD's contribution and quality through evaluations across three tasks: open-set aerial detection, open-vocabulary aerial detection (OVAD), and remote sensing visual grounding (RSVG).
> > >
> > > - **Open-set aerial detection:** Our benchmark, based on MI-OAD, demonstrates that models trained on MI-OAD effectively handle open-set (vocabulary, phrase, and sentence-input) detection tasks.
> > > - **OVAD task:** As detailed in Table 2 of our rebuttal, integrating MI-OAD into pre-training notably boosts LAE-DINO performance across prominent open-vocabulary aerial detection benchmarks (e.g., DIOR, DOTA v2.0, and LAE-80C).
> > > - **RSVG task:** Supplementary Tables 4 and 5 illustrate significant performance improvements in RSVG tasks when pretraining with MI-OAD, achieving state-of-the-art results on DIOR-RSVG and OPT-RSVG datasets.
> > >
> > > ### Selection of Comparison Methods:
> > >
> > > Given that open-set aerial detection remains an emerging area, comparison methods are limited. Therefore:
> > >
> > > - We initially selected two prominent, representative natural-image open-set detectors: **Grounding DINO [ECCV'24]** and **YOLO-World [CVPR'24]**.
> > > - We further strengthened our evaluation by including the latest state-of-the-art open-vocabulary aerial detection method, **LAE-DINO [AAAI’25]**, and assessed performance on its established open-vocabulary aerial detection benchmarks.
> > > - Additionally, we evaluated Grounding DINO (pretrained on MI-OAD) against prominent RSVG methods (including **MGVLF [TGARS’23], LPVA [TGARS’24]**, and seven other competing methods) on OPT-RSVG and DIOR-RSVG, as detailed in Supplementary Tables 4 and 5.
> > >
> > > We sincerely appreciate your insights and welcome any further detailed suggestions you might have regarding our evaluation. We remain committed to enhancing the quality and impact of our work based on your valuable feedback.

---

### Official Review · Reviewer_MAC6 · 2025-07-01

**Rating:** 4
**Confidence:** 4

**Summary:**

This paper introduces a new benchmark dataset, MI-OAD, for open-set object detection in aerial imagery. To address the scarcity of large-scale, semantically rich grounding data in the aerial domain, the authors propose an automated annotation pipeline called the OS-W2S Label Engine. This pipeline leverages a vision-language model to generate multi-level textual descriptions (word, phrase, sentence) for object instances in aerial images.

**Dataset Code Accessibility:**

Partly

**Dataset Code Comments:**

The main contribution of this paper is an automated annotation pipeline. However, the provided code does not include the claimed annotation process described in the paper, which only contains the YOLO-World and MMDetection libraries.

**Ethical Considerations:**

No, there are no or only very minor ethics concerns

**Final Justification:**

Some of my concerns regarding the writing and the use of experimental models have been addressed, and I have decided to raise my score.

**Limitations Weaknesses:**

1. This paper’s writing quality falls short of the standard expected for acceptance of NeurIPS. For example, the discussion of prior dataset limitations in the introduction is not sufficiently clear. It also lacks a clear explanation of how the small scale of existing datasets has constrained the development of open-set object detection. In addition, the authors are strongly encouraged to include a subsection detailing the implementation. Some critical experiments are strongly recommended to be moved from the supplementary material to the main paper, such as Comparison with existing remote-sensing grounding datasets and Comparison with state-of-the-art methods on the OPT-RSVG test set.

2. I have some concerns regarding the quality of the dataset. Compared to planar images, aerial imagery, despite post-processing, may still suffers from certain distortions. The authors use a VLM, InternVL-2.5-38B-AWQ, primarily trained on planar images to annotate these aerial images, which raises doubts about the resulting dataset’s quality. Additionally, the authors are encouraged to compare the annotation results across multiple VLMs, since each model has its own biases, and it is difficult to determine which one performs best for annotating aerial imagery.

**Strengths Contributions:**

1. The OS-W2S Label Engine is a well-designed, automated pipeline that produces fine-grained, multi-instance annotations with rich semantic content, enabling practical open-set detection in complex aerial scenes.

2. The scale of the dataset is commendable, surpassing previous datasets in both the number of image–caption pairs and the diversity of object categories.

3. The authors provide a thorough description of their OS-W2S Label Engine, which is used for data annotation, and they also discuss potential limitations of their work at the end of the paper.

---

> ### Author Rebuttal · Authors · 2025-07-31
>
> Thank you for your valuable feedback.
>
> ### **Response to Comment 1:**
>
> **(1) Clarification on Prior Dataset Limitations:**
>
> Existing dataset limitations fall into four major points. Limited by space, our initial introduction (lines 43–58) provided a brief overview. Following your suggestion, a detialed overview will be added in the revised version:
>
> - **Lack of Scene Diversity:** Existing RSVG datasets predominantly rely on the DIOR dataset and apply restrictive conditions, such as limiting images to fewer than five objects per category. Although such constraints help ensure annotation quality, they also exclude complex scenes that are essential for training robust open-set detectors.
> - **Limited Caption Diversity:** Current RSVG datasets predominantly employ fixed templates and rule-based attribute extraction. For example, relative position attributes are derived solely from bounding-box annotations to describe relationships between instances, neglecting richer contextual information regarding the surrounding environment.
> - **Single-Instance Annotation:** Existing RSVG datasets focus exclusively on referring grounding tasks, associating one caption with a single instance. However, practical applications often require models to retrieve all instances matching a given caption. The number of retrieved instances should be determined by the specificity of the caption, rather than always being limited to a single result.
> - **Limited Dataset Scale:** The largest available RSVG dataset comprises approximately 562,452 images and 48,952 image-caption pairs. Compared to successful natural-image-based open-set detectors (e.g., Grounding DINO v1.5 with 20M grounding data, DINO-X with 100M grounding data), existing aerial RSVG datasets are severely limited in scale. This substantial gap in data scale severely restricts the development of robust open-set aerial detection models.
>
> **(2) Impact of Small Scale:**
>
> The impact of small scale were briefly discussed in the related work section (lines 136–141). Additionally, following your suggestion, we conducted supplementary experiments that further demonstrate how insufficient dataset scale constrains open-set detection performance. Due to space limitations, these additional results will be included in the supplementary material.
>
> The impact of small scale has been relatively thoroughly validated by combining Table 1 (tested on MI-OAD) and Table 2 (tested on DIOR, DOTA v2.0, and LAE-80C).
>
> - **Experiment 1:** We compared Grounding DINO pretrained on the smaller OPT-RSVG dataset to pretraining on the MI-OAD FT-Set. As shown in Table 1, pretraining on the small-scale dataset led to substantially lower performance, clearly demonstrating the adverse impact of limited data scale on model generalization.
> - **Experiment 2:** We also evaluated LAE-DINO pretrained on very recently released LAE-1M (currently the largest open-vocabulary aerial detection dataset, containing over 1,600 categories) in comparison with MI-OAD FT-Set.  The results further underscore limited textual diversity as a critical shortcoming of existing datasets.
>
> Consequently, a larger, caption-diverse dataset is urgently needed to advance open-set aerial detection.
>
> **Table 1. Comparison on MI-OAD Val-FT Set:**
>
> | Method         | Pre-Training Data | Detection AP₅₀ | Detection R@100 | Phrase AP₅₀ | Phrase R@1 | Phrase R@10 | Phrase R@100 | Sentence AP₅₀ | Sentence R@1 | Sentence R@10 | Sentence R@100 |
> | -------------- | ----------------- | -------------- | --------------- | ----------- | ---------- | ----------- | ------------ | ------------- | ------------ | ------------- | -------------- |
> | Grounding DINO | OPT-RSVG          | 3.3            | 35.4            | 9.7         | 15.0       | 33.3        | 34.2         | 7.0           | 14.5         | 26.1          | 26.4           |
> | Grounding DINO | MI-OAD            | 37.1           | 70.1            | 57.8        | 35.2       | 74.4        | 91.5         | 56.4          | 44.1         | 78.0          | 90.3           |
> | LAE-DINO       | LAE-1M            | 24.8           | 70.1            | 22.9        | 22.9       | 56.0        | 79.7         | 5.1           | 17.5         | 47.1          | 67.9           |
> | LAE-DINO       | MI-OAD            | 40.0           | 75.0            | 59.4        | 35.8       | 74.9        | 92.6         | 57.8          | 44.5         | 78.9          | 92.4           |
>
> **(3) Detailed Implementation:**
>
> Section 3.2 of our main text provides a comprehensive description of the OS-W2S pipeline, and supplementary Section C.1 offers extensive experimental details. However, we acknowledge additional implementation clarity could benefit reproducibility.
>
> **Action:** We will enhance our GitHub repository by including fully commented code and a clearly structured README to improve transparency and reproducibility.
>
> **(4) Critical Experiment Integration:**
>
> We fully agree with your suggestion. Initially, critical experiments comparing existing grounding datasets and state-of-the-art methods were placed in supplementary materials due to page limitations.
>
> **Action:** In the final version, we will move key comparative experiments into the main text. Specifically, we will concisely summarize experiments that clearly demonstrate MI-OAD’s superiority in RSVG and open-vocabulary aerial detection tasks. Comprehensive results will remain accessible in the supplementary material, balancing clarity and completeness within page constraints.
>
> ### **Response to Comment 2:**
>
> **(1) Minimizing Domain Gap**
>
> We acknowledge that using LVLMs trained primarily on planar images to annotate aerial data may introduce a domain gap. To mitigate this, we have meticulously designed the data pre-processing pipeline in the OS-W2S Label Engine. Specifically, we extract each instance region and its corresponding foreground based on bounding box annotations, and generate three prior attributes (category, size, and absolute position) using carefully crafted rules. These steps provide the LVLM with rich, context-aware prompts, enabling it to generate precise, instance-focused annotations.
>
> - **Experimental Validation**
>
>   Extensive experiments and qualitative analyses further demonstrate the effectiveness of MI-OAD:
>
>   - Improvements in Remote Sensing Visual Grounding: Supplementary Tables 4 and 5 show that pretraining on MI-OAD significantly enhances performance in RSVG tasks, achieving state-of-the-art results on both DIOR-RSVG and OPT-RSVG.
>
>   - Improvements in Open-Vocabulary Aerial Detection:  We conducted additional experiments with the LAE-DINO model, which further validate the effectiveness of MI-OAD. As shown in Table 2, incorporating MI-OAD during pre-training notably improves performance across several open-vocabulary aerial detection benchmarks.
>
>     **Table 2. Open-Vocabulary detection performance on DIOR, DOTAv2.0, and LAE-80C benchmarks.**
>
>     | Training Status | Method   | Pre-Training Data | DIOR AP₅₀ | DOTAv2.0 mAP | LAE-80C mAP |
>     | --------------- | -------- | ----------------- | --------- | ------------ | ----------- |
>     | Fully Trained   | LAE-DINO | LAE-1M            | 85.5      | 46.8         | 20.2        |
>     | Training        | LAE-DINO | LAE-1M            | 83.4      | 45.3         | 17.8        |
>     | Training        | LAE-DINO | + MI-OAD          | 91.6      | 50.3         | 20.5        |
>
>     **Note:** “Fully Trained” results reflect the original reported metrics, where the model was thoroughly trained for 32 epochs with 4×A100 GPUs. Due to time constraints, the “Training” results are from models trained for 13 epochs with 32 RTX 4090 GPUs; thus, these models were not fully converged. Nevertheless, after incorporating the MI-OAD dataset, the performance already surpasses the “Fully Trained” baseline, further demonstrating the effectiveness of our dataset.
>
> **(2) Rationale for Selecting InternVL-2.5-38B-AWQ:**
>
> The quality of annotation is significantly depends on the capabilities of the selected LVLM. While larger models typically achieve higher annotation quality, they also demand greater computational resources.
>
> - **Initial Model Evaluation:** During our preliminary assessment, we evaluated several leading LVLMs (e.g., Qwen2-VL, InternVL2.5 series). At the time of dataset construction, InternVL2.5 was the strongest open-source LVLM available. Among all the models tested, only Qwen2-VL-72B, InternVL2.5-78B, and InternVL-2.5-38B (including their quantized versions) consistently achieved a **100% template-parsing success rate** as verified by regular expression checks, demonstrating their ability to fully understand and reliably follow our output format.
>
> - **Efficiency and Cost Analysis.** InternVL-2.5-78B requires a minimum of **four 80GB GPUs** (4×A100), while InternVL-2.5-38B-AWQ can operate on **eight 24GB GPUs** (8×RTX 4090). In our trials, annotating 100 identical images took approximately **50 minutes** for both configurations. Considering typical GPU rental prices (`4×A100 ≈ $4.18/h vs. 8×RTX4090 ≈ $2.09/h` ), InternVL-2.5-38B-AWQ reduces hardware costs by about half, thereby improving accessibility and reproducibility for the research community. Based on this balance of efficiency, cost, and scalability, we ultimately selected InternVL-2.5-38B-AWQ for our large-scale annotation.
>
> **Flexibility:** The OS-W2S Label Engine is model-agnostic and supports seamless integration of future LVLM advancements, including emerging remote sensing–specialized LVLMs.
>
> **Action:** A detailed subsection outlining our LVLM selection criteria, as well as cross-VLM comparison results, will be included in the supplementary.
>
> ### **Response to Dataset Code Comments:**
>
> Considering the Datasets & Benchmarks Track, our initial submission emphasized releasing the complete dataset and evaluation code. We reaffirm our commitment to **fully open-sourcing** the dataset, annotation pipeline code, and all benchmarking scripts for the community.
>
> ---

---

> > ### Comment · Reviewer_MAC6 · 2025-08-02
> >
> > I would like to thank the authors for their detailed response regarding the limitations of prior datasets and hopes that the authors can summarize these limitations and include them in the revised Introduction.
> >
> > In addition, I still finds it unclear what specific benefits the proposed pre-processing brings to help the pre-trained LVM better understand distorted images, and asks the authors to provide further explanation. Moreover, I does not believe that model size has an absolute correlation with performance on a specific task. If possible, the authors are encouraged to conduct an experiment to verify that the InternVL-2.5-38B-AWQ model they used is indeed suitable for annotating aerial images.
> >
> > I would like to raise the score if the authors are able to address these concerns.

---

> > > ### Author Response · Authors · 2025-08-04
> > > **(1/3) Clarification on the Effectiveness of Proposed Pre-processing**
> > >
> > > Dear Reviewer, thank you for your valuable feedback and continued engagement. And we fully understand your concerns and provide detailed clarifications below:
> > >
> > > > **Why is the proposed pre-processing effective for LVLMs?**
> > >
> > > As the reviewer correctly pointed out, the LVLM we employed was primarily trained on planar images, resulting in a domain gap when annotating aerial imagery. To effectively bridge this gap, we carefully designed the pre-processing pipeline in the OS-W2S Label Engine based on two guiding principles:
> > >
> > > - **Ensuring Attribute Correctness through Reliable Priors:**
> > >
> > >   The quality of our annotations heavily depends on the accuracy of various instance attributes (category, size, color, geometry and relative/abosolute position). To guarantee the correctness of these attributes and thus enhance annotation quality, we explicitly provide the LVLM with three **prior attributes** per instance:
> > >
> > >   - **Category**: Deterministically obtained from object detection annotations.
> > >   - **Size and Absolute Location**: Computed using deterministic, rule-based methods from bounding boxes—for example, size is calculated as the area ratio, and absolute location is determined by the instance’s coordinates within the image.
> > >
> > >   Providing these prior attributes helps alleviate annotation errors induced by domain gap and reduces ambiguity, thereby improving the LVLM's ability to generate reliable annotations.
> > >
> > > - **Maximizing Informative Visual Input:**
> > >
> > >   Appropriate cropping helps the LVLM focus on the correct target and provide more fine-grained information. However, excessive zooming may omit crucial contextual information, while insufficient zooming may result in missing important object details. After iterative experimentation, we adopted a two-stage cropping strategy to optimally balance fine details and relevant context:
> > >
> > >   - **Instance Region Cropping (Stage 1):** Each object is initially cropped based on its bounding box and enlarged by a scale factor, as shown in Figure 2 of the main text. Alongside this cropped region, we provide the known **category and size** priors. This approach ensures the LVLM concentrates on the specific instance, facilitating accurate generation of **color** and **geometry** attributes while minimizing potential misclassifications or hallucinations.
> > >   - **Foreground Region Extraction (Stage 2):** Subsequently, we generate a slightly expanded **foreground crop** using a foreground-extraction algorithm based on bounding boxes. This enlarged crop includes the immediate surroundings of the target object. The target remains highlighted with red box to guide model knows exactly which object to describe. Providing this local contextual information enables the LVLM to accurately infer **relative position** attributes, maintaining a clear focus and reducing confusion caused by broader background noise.
> > >
> > > **Summary :**  Our carefully structured pre-processing strategy effectively addresses domain gaps by:
> > >
> > > - Utilizing **instance-level crops** to guide the LVLM's focus on fine-grained details.
> > > - Employing **foreground-level crops** to provide essential local context without introducing excessive background noise.
> > > - Integrating reliable **prior attribute extraction** for attributes that are most sensitive to the domain gap (i.e., category, size, and absolute location), ensuring their accuracy by leveraging detection annotation priors. For the remaining attributes (color, geometry, and relative position), which are relatively robust to domain shifts, the LVLM is capable of providing accurate annotations. This combination further mitigates the difficulties arising from the domain gap inherent to aerial imagery.
> > >
> > > The effectiveness of this approach aligns with recent research findings. For example, DAM [1] proposes that global image features alone may obscure important regional details, while naive zoom-in often discard essential context. They propose a dual-branch architecture that simultaneously captures fine-grained detail and global context. Similarly, RegionGPT [2] and Alpha-CLIP [3] leverage region-aware input strategies to enhance localized understanding.
> > >
> > > ------
> > >
> > > **References:**
> > >
> > > [1] Lian L, Ding Y, Ge Y, et al. Describe anything: Detailed localized image and video captioning[J]. arXiv preprint arXiv:2504.16072, 2025.
> > >
> > > [2] Guo Q, De Mello S, Yin H, et al. Regiongpt: Towards region understanding vision language model[C]//Proceedings of the IEEE/CVF Conference on Computer Vision and Pattern Recognition. 2024: 13796-13806.
> > >
> > > [3] Sun Z, Fang Y, Wu T, et al. Alpha-clip: A clip model focusing on wherever you want[C]//Proceedings of the IEEE/CVF conference on computer vision and pattern recognition. 2024: 13019-13029.
> > >
> > > ---

---

> > > ### Author Response · Authors · 2025-08-04
> > > **(2/3) Clarification on Impact of Model Size for Annotation Quality**
> > >
> > > > **Impact of Model Size on Annotation Quality**
> > >
> > > We agree with the reviewer that a larger model size does not necessarily guarantee better performance for every specific domain or task. However, in our specific scenario, the images represent a domain shift, while textual caption annotation primarily relies on the LLM’s ability to comprehend and interpret carefully designed prompts, a capacity correlated with model size according to current LLM research [1-3]. Specifically, selecting an appropriate LVLM for aerial image annotation involves several practical considerations:
> > >
> > > - **Vision Branch Capabilities:** As discussed previously, our pre-processing strategy provides detailed, fine-grained instance regions along with three explicit prior attributes, effectively mitigating the domain gap in visual processing.
> > > - **Text Branch Requirements:** To enable the LLM to effectively utilize prior knowledge and generate multi-level descriptions—from individual words and phrases to complete sentences—we meticulously crafted a set of structured prompts. A larger LLM has a superior capability to accurately interpret these prompts, strictly adhere to their designed instructions, and generate semantically rich and coherent caption annotations. Therefore, the inherent complexity of our task necessitates that the LLM component should not be excessively small.
> > > - **Efficiency and Practicality:** The chosen model must strike a balance between annotation quality and computational efficiency, ensuring practical and reproducible large-scale annotation. Given that an 8-GPU RTX 4090 workstation represents commonly accessible hardware, we chose this configuration as our baseline for model selection.
> > >
> > > After extensive comparative evaluation, we selected InternVL-2.5-38B-AWQ as the most suitable model meeting these criteria.
> > >
> > > ---
> > >
> > > **References:**
> > >
> > >  [1] Kaplan J, McCandlish S, Henighan T, et al. Scaling laws for neural language models[J]. arXiv preprint arXiv:2001.08361, 2020.
> > >
> > > [2] Rae J W, Borgeaud S, Cai T, et al. Scaling language models: Methods, analysis & insights from training gopher[J]. arXiv preprint arXiv:2112.11446, 2021.
> > >
> > > [3] Chowdhery A, Narang S, Devlin J, et al. Palm: Scaling language modeling with pathways[J]. Journal of Machine Learning Research, 2023, 24(240): 1-113.
> > >
> > > ------

---

> > > ### Author Response · Authors · 2025-08-05
> > >
> > > Dear Reviewer MAC6,
> > >
> > > Thank you again for your great efforts and valuable comments. We hope our detailed responses have addressed all your concerns.
> > >
> > > As the discussion phase is nearing its end, we are looking forward to any additional feedback you might have. If you have any further questions or concerns, we would be happy to provide additional clarification.
> > >
> > > Best regards,
> > >
> > > The Authors

---

> > > > ### Comment · Reviewer_MAC6 · 2025-08-06
> > > >
> > > > Thanks for the detailed response, which has addressed some of my concerns. I will raise my score.

---

> > > > > ### Author Response · Authors · 2025-08-06
> > > > >
> > > > > Thank you for reviewing our response and raising your rating. We sincerely appreciate your time and effort.
> > > > >
> > > > > Best regards,
> > > > >
> > > > > The Authors

---

> > ### Author Response · Authors · 2025-08-04
> > **(3/3) Validation of InternVL-2.5-38B-AWQ for Aerial Image Annotation**
> >
> > > **Evidence Supporting InternVL-2.5-38B-AWQ's Capability for Annotating Aerial Images**
> >
> > Following the reviewer's suggestion, we provide the following evidence:
> >
> > - **Training Data Coverage:** InternVL2.5 [1] was pre-trained on 16.3 million image-text pairs sourced from large-scale web datasets, including OpenImages-Caption [2], InternVL-SA-1B-Caption [3], ShareGPT4V [4], and LAION-5B [5]. Recent research by LAION-EO [6] demonstrates that such datasets include amounts of aerial imagery, suggesting that InternVL2.5 has certain generalization capabilities for aerial scenes.
> >
> > - **Automatic Cross-Model Validation:** To rigorously evaluate annotation quality, we sampled 300 images (~1,765 instances) from the MI-OAD dataset. Two powerful independent models—a stronger open-source LVLM (InternVL3-78B) and a leading closed-source model (GPT-4o-mini)—were employed to validate annotations for color, geometry (using instance crops), and relative position (using foreground crops). The results are as follows:
> >
> >   - InternVL3-78B: color (98.98%), geometry (99.21%), relative position (97.90%)
> >   - GPT-4o-mini: color (96.88%), geometry (94.39%), relative position (96.92%)
> >
> >   These metrics confirm that InternVL-2.5-38B-AWQ reliably produces accurate annotations for aerial imagery.
> >
> > - **Adoption in Recent Studies:** The capabilities of using LVLMs primarily trained on planar images for annotating aerial images has been validated by recent works. For example, LAE-DINO [7] employed InternVL-1.5 [8] to construct large-scale open-vocabulary aerial datasets. EarthDial [9] utilized InternLM‑XComposer2‑VL [10] for generating aerial question-answer pairs, while RS5M [11] leveraged BLIP-2 [12] for caption generation from category-only aerial data. These studies collectively illustrate the proven applicability and increasing adoption of planar-trained LVLMs within the remote sensing research community.
> >
> > ------
> >
> > **References:**
> >
> > [1] Chen Z, Wang W, Cao Y, et al. Expanding performance boundaries of open-source multimodal models with model, data, and test-time scaling[J]. arXiv preprint arXiv:2412.05271, 2024.
> >
> > [2] Kuznetsova A, Rom H, Alldrin N, et al. The open images dataset v4: Unified image classification, object detection, and visual relationship detection at scale[J]. International journal of computer vision, 2020.
> >
> > [3] Kirillov A, Mintun E, Ravi N, et al. Segment anything[C]//Proceedings of the IEEE/CVF international conference on computer vision. 2023.
> >
> > [4] Chen L, Li J, Dong X, et al. Sharegpt4v: Improving large multi-modal models with better captions[C]//European Conference on Computer Vision, 2024.
> >
> > [5] Schuhmann C, Beaumont R, Vencu R, et al. Laion-5b: An open large-scale dataset for training next generation image-text models[J]. Advances in neural information processing systems, 2022.
> >
> > [6] Czerkawski M, Francis A. From LAION-5B to LAION-EO: Filtering billions of images using anchor datasets for satellite image extraction[J]. arXiv preprint arXiv:2309.15535, 2023.
> >
> > [7] Pan J, Liu Y, Fu Y, et al. Locate anything on earth: Advancing open-vocabulary object detection for remote sensing community[C]//Proceedings of the AAAI Conference on Artificial Intelligence. 2025.
> >
> > [8] Chen Z, Wang W, Tian H, et al. How far are we to gpt-4v? closing the gap to commercial multimodal models with open-source suites[J]. Science China Information Sciences, 2024.
> >
> > [9] Soni S, Dudhane A, Debary H, et al. Earthdial: Turning multi-sensory earth observations to interactive dialogues[C]//Proceedings of the Computer Vision and Pattern Recognition Conference. 2025.
> >
> > [10] Dong X, Zhang P, Zang Y, et al. Internlm-xcomposer2: Mastering free-form text-image composition and comprehension in vision-language large model[J]. arXiv preprint arXiv:2401.16420, 2024.
> >
> > [11] Zhang Z, Zhao T, Guo Y, et al. Rs5m and georsclip: A large scale vision-language dataset and a large vision-language model for remote sensing[J]. IEEE Transactions on Geoscience and Remote Sensing, 2024.
> >
> > [12] Li J, Li D, Savarese S, et al. Blip-2: Bootstrapping language-image pre-training with frozen image encoders and large language models[C]//International conference on machine learning. PMLR, 2023.

---

### Official Review · Reviewer_1aRU · 2025-07-03

**Rating:** 5
**Confidence:** 5

**Summary:**

This paper presents MI-OAD, a large-scale, multi-instance, open-set aerial object detection dataset, along with OS-W2S Label Engine, an automatic annotation pipeline for generating rich textual annotations (words, phrases, and sentences) for aerial images. This paper further demonstrate, through comprehensive experiments, that models fine-tuned on MI-OAD (e.g. Grounding DINO) achieve significantly improved open-set aerial detection and grounding performance, especially under zero-shot conditions.

**Dataset Code Accessibility:**

Yes

**Ethical Considerations:**

No, there are no or only very minor ethics concerns

**Final Justification:**

The authors have made a great effort to build the dataset and addressed my primary concerns, so I decided to raise my score.

**Limitations Weaknesses:**

1.Regarding data sources: The authors expanded semantic annotations based on existing datasets such as Xview, DOTA, and DIOR, but did not leverage tools like OSM or GEE to acquire a larger-scale, more diverse image corpus. This weakens the contribution of the work to the field, as image diversity beyond semantic diversity, is a crucial factor directly affecting a model’s generalization ability on unseen images.
2. Insufficient model evaluation: Existing open-vocabulary aerial object detection models, such as LAE-DINO, were not included in the evaluation. As a result, the authors lack stronger evidence to substantiate that their proposed dataset is a superior benchmark. The authors should compare against the benchmark proposed by LAE-DINO and demonstrate clear advantages in open-vocabulary detection performance.

**Strengths Contributions:**

1. Significance and Novelty
This paper tackles a clear and important gap in language-guided open-set aerial object detection: the severe scarcity of large-scale, diverse grounding datasets for remote sensing imagery. While open-set detection has advanced considerably in natural image domains, aerial object detection remains limited due to small, template-based captioning datasets. The proposed MI-OAD dataset represents a substantial leap forward, with 2M image-caption pairs, achieving 40x the size of existing aerial grounding datasets.
2. Technical Innovation
The proposed OS-W2S Label Engine is a well-designed, automatic, multi-stage pipeline combining vision-language models and structured attribute-based caption generation. Its design ensures scene diversity, caption diversity, and multi-instance annotation, explicitly overcoming the limitations of existing datasets.
3. Benchmark Establishment and Impact
By establishing a standardized evaluation protocol for open-set aerial detection at vocabulary, phrase, and sentence levels (Section 4.3), the paper positions MI-OAD as a foundational benchmark for future research. The experiments demonstrate that models trained or domain-adapted on MI-OAD (e.g., Grounding DINO) substantially outperform baselines under zero-shot and fine-tuned settings (Table 1, p.8–9), with a notable +31.1 AP50 improvement in sentence-level grounding.

---

> ### Author Rebuttal · Authors · 2025-07-31
>
> We greatly appreciate your detailed feedback and recognition of our contributions. We hope our response below effectively addresses your concerns.
>
> > Comment 1: Regarding data sources: The authors expanded semantic annotations based on existing datasets such as Xview, DOTA, and DIOR, but did not leverage tools like OSM or GEE to acquire a larger-scale, more diverse image corpus. This weakens the contribution of the work to the field, as image diversity beyond semantic diversity, is a crucial factor directly affecting a model’s generalization ability on unseen images.
>
> Thank you for this insightful comment. We fully agree that both image diversity and semantic diversity are critical for advancing open-set aerial detection. In constructing MI-OAD, we carefully balanced the semantic richness of annotations with the diversity and quality of the available images.
>
> - **On the semantic side**, we leveraged LVLMs through our proposed Label Engine pipeline to generate rich, diverse captions and fine-grained attribute annotations for each object. This significantly enhances the semantic content of MI-OAD beyond what existing datasets provide.
> - **On the image side**, we aimed to maximize the diversity of scenes and targets included, while also ensuring high-precision bounding box annotations for all objects. To achieve this, we built MI-OAD on eight widely used, high-quality aerial object detection datasets (e.g., xView, DOTA, DIOR). This choice guarantees broad category coverage and provides accurate, human-verified bounding boxes for every object.
>
> The reviewer’s suggestion to incorporate **GEE and OSM** data is indeed insightful and points to a promising direction for further increasing image diversity in future work. We agree that tapping into these resources could meaningfully enhance the diversity of our dataset. Below, we would like to clarify why we did not initially include OSM/GEE data in the current version of MI-OAD:
>
> - **GEE Data:** Images fetched via GEE (typically by specifying geographic coordinates) come with only a single location coordinate and lack any object-specific annotations (no bounding boxes or category labels). Therefore, GEE data can only serve as a form of weak supervision.
> - **OSM Data:** OSM offers polygon outlines for certain large-scale structures (e.g., buildings, land parcels, lakes), but it does not cover many of the fine-grained object categories that are critical to aerial object detection (such as various vehicles, aircraft, pedestrians, etc.).
>
> Nevertheless, we strongly agree that expanding image source diversity is a crucial factor that directly affects a model’s generalization ability on unseen data. Incorporating GEE and OSM data is a forward-looking idea, one well worth exploring in the future. For instance, OSM could help gather additional samples for object categories or scene types that are underrepresented in our current dataset, and GEE could provide satellite images from novel geographic regions or unique environments not covered by existing datasets. We could leverage the metadata from OSM/GEE as coarse initial labels and then design a  VLM-assisted Label Engine to refine these labels and generate precise bounding boxes. Finally, we would utilize our OS-W2S Label Engine to augment the caption annotations. This pipeline would further address the image and semantic diversity challenges in open-set aerial object detection.
>
> **Summary:** In terms of data source selection, we aimed to cover as many relevant aerial object classes as possible while ensuring precise bounding boxes. Ultimately, we decided to select existing detection datasets as image sources. We appreciate the reviewer’s suggestion and will certainly consider incorporating broader image sources such as OSM and GEE in future expansions of MI-OAD, with the goal of contributing an even more valuable dataset to the open-set aerial detection community.
>
> ---
>
> > Comment 2: Insufficient model evaluation: Existing open-vocabulary aerial object detection models, such as LAE-DINO, were not included in the evaluation. As a result, the authors lack stronger evidence to substantiate that their proposed dataset is a superior benchmark. The authors should compare against the benchmark proposed by LAE-DINO and demonstrate clear advantages in open-vocabulary detection performance.
>
> Thank you for bringing this up. Below, we sequentially (1) clarify why LAE-DINO was not initially evaluated on the MI-OAD dataset, and (2) present additional experiments demonstrating clear advantages of the MI-OAD dataset for open-vocabulary aerial detection tasks.
>
> - **Reasons for Excluding LAE-DINO from Initial Evaluation**
>
> In our initial evaluation, we employed a rigorous Base/Novel class split aimed at clearly assessing the zero-shot transfer performance. Existing open-vocabulary aerial object detection models, such as LAE-DINO [1], do not adopt this rigorous Base/Novel class distinction. Directly evaluating LAE-DINO on our benchmark would lead to information leakage for Novel classes since the LAE-1M dataset contains categories that overlap with MI-OAD’s Novel classes. As demonstrated in Table 1, the leakage results in inflated performance metrics for LAE-DINO in Detection and Phrase Grounding tasks. However, its performance on the more challenging Sentence Grounding task, which provides richer contextual information, remains comparatively lower. Consequently, directly including LAE-DINO in our benchmark comparison would be unfair to other methods.
>
> **Table 1. Performance comparison of representative methods on the MI-OAD dataset (Zero-shot transfer with novel classes, without domain adaptation).**
>
> | Method             | Detection AP₅₀ | Detection R@100 | Phrase AP₅₀ | Phrase R@1 | Phrase R@10 | Phrase R@100 | Sentence AP₅₀ | Sentence R@1 | Sentence R@10 | Sentence R@100 |
> | ------------------ | -------------- | --------------- | ----------- | ---------- | ----------- | ------------ | ------------- | ------------ | ------------- | -------------- |
> | Yolo-World [2]     | 3.2            | 37.1            | 3.8         | 6.8        | 25.0        | 34.4         | 1.4           | 4.3          | 16.9          | 24.6           |
> | Grounding DINO [3] | 4.0            | 49.6            | 9.2         | 10.7       | 35.1        | 50.4         | 5.2           | 10.3         | 33.8          | 42.9           |
> | LAE-DINO [1]       | 28.7           | 68.5            | 24.4        | 22.5       | 53.4        | 76.2         | 7.8           | 18.5         | 46.5          | 67.0           |
>
> - **Performance Enhancement on Open-Vocabulary Aerial Detection with MI-OAD**
>
> Following your valuable suggestion, we conducted additional experiments to validate the advantages of MI-OAD for open-vocabulary aerial detection tasks. As illustrated in Table 2, incorporating MI-OAD notably enhances model performance across various open-vocabualry aerial detection benchmarks.
>
> **Table 2. Open-Vocabulary aerial detection performance on DIOR, DOTAv2.0, and LAE-80C benchmarks.**
>
> | Training Status | Method   | Pre-Training Data | DIOR AP₅₀ | DOTAv2.0 mAP | LAE-80C mAP |
> | --------------- | -------- | ----------------- | --------- | ------------ | ----------- |
> | Fully Trained   | LAE-DINO | LAE-1M            | 85.5      | 46.8         | 20.2        |
> | Training        | LAE-DINO | LAE-1M            | 83.4      | 45.3         | 17.8        |
> | Training        | LAE-DINO | + MI-OAD          | 91.6      | 50.3         | 20.5        |
>
> **Note:** “Fully Trained” results reflect the original reported metrics, where the model was thoroughly trained for 32 epochs with 4×A100 GPUs. Due to time constraints, the “Training” results are from models trained for 13 epochs with 32 RTX 4090 GPUs; thus, these models were not fully converged. Nevertheless, after incorporating the MI-OAD dataset, the performance already surpasses the “Fully Trained” baseline, further demonstrating the effectiveness of our dataset.
>
> **Action:** Following your suggestion, we will report the LAE-DINO benchmark results separately from Table 1 in the main paper to ensure a fair comparison. Additionally, we will include comprehensive experimental results using LAE-DINO as a baseline on open-vocabulary detection benchmarks in the final version, further demonstrating the effectiveness and generalizability of the MI-OAD dataset.
>
> ---
>
> **References**
>
> [1] Pan J, Liu Y, Fu Y, et al. Locate anything on earth: Advancing open-vocabulary object detection for remote sensing community[C]//Proceedings of the AAAI Conference on Artificial Intelligence. 2025, 39(6): 6281-6289.
>
> [2] Cheng T, Song L, Ge Y, et al. Yolo-world: Real-time open-vocabulary object detection[C]//Proceedings of the IEEE/CVF conference on computer vision and pattern recognition. 2024: 16901-16911.
>
> [3] Liu S, Zeng Z, Ren T, et al. Grounding dino: Marrying dino with grounded pre-training for open-set object detection[C]//European conference on computer vision. 2024: 38-55.
>
> ---

---

### Official Review · Reviewer_KjQx · 2025-07-11

**Rating:** 4
**Confidence:** 4

**Summary:**

This work concentrates on benchmarking within the field of Open Vocabulary Aerial Detection. To address the current shortage of large-scale benchmarks in this domain, it introduces a VLM-based annotation engine named OS-W2S. Leveraging this engine, the study integrates eight datasets to create a large-scale benchmark, MI-OAD, which encompasses diverse scenarios and provides fine-grained, multi-level annotations at the vocabulary, phrase, and sentence levels.

**Dataset Code Accessibility:**

Partly

**Dataset Code Comments:**

The authors have committed to open-sourcing the dataset in the future. Possibly due to the requirements of blind review, the authors did not directly provide a GitHub link; however, the project homepage linked to this paper (MI-OAD) can be found on GitHub, where the code for the evaluation part has already been open-sourced.

**Ethical Comments:**

The images in the dataset proposed by this work are all sourced from existing widely-used datasets. Only a Vision-Language Model (VLM) was introduced for re-annotation, and no ethical issues are involved.

**Ethical Considerations:**

No, there are no or only very minor ethics concerns

**Final Justification:**

After reading the authors' feedback, I still have concerns on the significance of building an open-set aerial detection benchmark based on well-established, high-quality datasets. I thus keep my initial rating.

**Limitations Weaknesses:**

1.Lack of filtering and selection. As described in the main text and appendix, this work simply concatenates images from eight different datasets without further filtering. Some images (e.g., certain images from VisDrone) exhibit severe motion blur and target imaging issues (almost indistinguishable to the naked eye). The absence of filtering may limit the overall quality of the dataset.
2.Lack of large-scale quality control. The authors did not employ a method such as training a small model or fine-tuning a large model to automatically assess annotation quality. Only a portion of the data was manually checked, which could result in potential annotation errors or hallucinations. Additionally, the vocabulary used to describe positions—such as "middle" and "center"—may have similar meanings but refer to different locations (vertical vs. horizontal), potentially leading to model misjudgments. The authors did not investigate or address these issues

**Strengths Contributions:**

1. This paper introduces the first large-scale open-vocabulary aerial object detection dataset. The MI-OAD dataset proposed in this work contains on the order of 160k images and provides millions of multi-level image-caption pairs, offering significant potential value.
2. The authors propose a relatively well-designed annotation scheme. The annotations in this work combine handcrafted rule design, interaction with Vision-Language Models (VLM), and partial manual inspection. The final annotations include multi-level descriptions (words, phrases, sentences) covering attributes and positional information
3. To ensure quality, the authors selected 1,000 images for review by five experts and manually curated 10,000 images to construct the test set.

---

> ### Author Rebuttal · Authors · 2025-07-31
>
> Thank you for your valuable time and constructive feedback! We hope to address your primary concerns below.
>
> > Comment 1: Lack of filtering and selection. As described in the main text and appendix, this work simply concatenates images from eight different datasets without further filtering. Some images (e.g., certain images from VisDrone) exhibit severe motion blur and target imaging issues (almost indistinguishable to the naked eye). The absence of filtering may limit the overall quality of the dataset.
>
> We understand your concern regarding potential quality issues arising from directly aggregating multiple datasets without additional filtering. To clarify our design choices, we elaborate on our initial decision from three perspectives:
>
> - **Dataset Selection:** We aim to construct an open-set aerial detection benchmark based on well-established, high-quality datasets. Thus, we carefully selected eight widely-used aerial object detection datasets, including VisDrone, DOTA, DIOR, among others. These datasets have become standardized benchmarks within the aerial detection community due to their high quality, diversity, and representativeness across various sensor platforms (e.g., UAVs, satellites) and image resolutions.
>
> - **Quality Control at Dataset Creation:** The selected datasets originally underwent rigorous human annotation procedures with comprehensive quality assurance mechanisms. Taking the xView [1] dataset as an example, it implemented a thorough three-stage quality control pipeline, including:
>
>   - Worker Quality Control: Annotators rotated as peer reviewers to identify and rectify errors related to categories, bounding box, and duplicate annotations.
>   - Supervisory Quality Control: Systematic identification and removal of duplicate entries, invalid labels/geometries.
>   - Expert Quality Control:  Experts created a gold-standard dataset by sampling and annotating six 1 km² image chips per batch. Worker annotations from batches representing 10%, 40%, 70%, and 100% completion stages were evaluated against this gold-standard. Batches only passed if they met or exceeded a precision of 0.75 and recall of 0.95 at IoU 0.5. Batches that failed expert quality control were remediated and resubmitted.
>
>   Given that these datasets were thoroughly vetted, we intentionally chose not to filter out challenging yet realistic samples, including images exhibiting motion blur. Retaining these samples preserves the authentic distributional characteristics encountered in real-world UAV and satellite scenarios, thereby enhancing the robustness of models trained on this data.
>
> - **Image Quality Annotation:**  **Following your suggestions, we employed Q-Align [2], a robust image quality assessment model, to evaluate each image in MI-OAD systematically.** Every image was assigned an Image Quality annotation, consisting of a numerical quality score and a categorical classification into five distinct levels: excellent, good, fair, poor, and bad. The distribution across these quality levels in MI-OAD is as follows: excellent (0.1%), good (33.0%), fair (43.1%), poor (21.7%), and bad (2.1%). Researchers can leverage these annotations to selectively filter the dataset based on their specific requirements. Both the assessment procedure and quality annotations will be made publicly available upon release.
>
> **Summary:** Our initial decision to include rare yet realistic blurred or challenging images supports the development of robust open-set detection methods. The inherent high quality of the selected datasets, combined with detailed image quality annotations, provides researchers with flexible and precise control over data filtering to meet diverse research objectives.
>
> ------
>
> > Comment 2:
> >
> > (1) Lack of large-scale quality control. The authors did not employ a method such as training a small model or fine-tuning a large model to automatically assess annotation quality. Only a portion of the data was manually checked, which could result in potential annotation errors or hallucinations.
> >
> > (2) Additionally, the vocabulary used to describe positions—such as "middle" and "center"—may have similar meanings but refer to different locations (vertical vs. horizontal), potentially leading to model misjudgments. The authors did not investigate or address these issues
>
> (1)  We appreciate your concern regarding large-scale annotation quality control. In the aerial detection domain, however, limited availability of large-scale, high-quality labeled data makes training or fine-tuning automated annotation quality models challenging. Aerial images typically feature dense distributions of small objects, further complicating automated quality assessment. Considering these constraints, we have meticulously designed the OS-W2S Label Engine with robust internal quality control mechanisms ensuring high annotation quality. We detail our quality control approach and validation procedures as follows:
>
> - **Quality Control within OS-W2S:**
>
>   - **Structured Prompts and Output Validation:** Through carefully crafted prompts, we enable the LVLM to fully comprehend task requirements and generate outputs in a predefined format. The outputs are further validated using regular expression matching, achieving a 100% parsing success rate.
>   - **Error and Hallucination Minimization:** We employed extensive preprocessing, including instance-region cropping, foreground extraction, and attribute priors from predefined rules, providing adequate context for accurate LVLM annotations (color, geometry, relative position).
>
> - **Benchmark Quality Assurance:** Rigorous manual reviews were conducted during test-set selection, guaranteeing MI-OAD as a reliable benchmark.
>
> - **Experimental Validation**
>
>   Extensive experiments and qualitative analyses further demonstrate the effectiveness of MI-OAD:
>
>   - Improvements in Remote Sensing Visual Grounding: Supplementary Tables 4 and 5 show that pretraining on MI-OAD significantly enhances performance in RSVG tasks, achieving state-of-the-art results on both DIOR-RSVG and OPT-RSVG.
>
>   - Improvements in Open-Vocabulary Aerial Detection:  We conducted additional experiments with the LAE-DINO model, which further validate the effectiveness of MI-OAD. As shown in Table 1, incorporating MI-OAD during pre-training notably improves performance across several open-vocabulary aerial detection benchmarks.
>
>     **Table 1. Open-Vocabulary aerial detection performance on DIOR, DOTAv2.0, and LAE-80C benchmarks.**
>
>     | Training Status | Method   | Pre-Training Data | DIOR AP₅₀ | DOTAv2.0 mAP | LAE-80C mAP |
>     | --------------- | -------- | ----------------- | --------- | ------------ | ----------- |
>     | Fully Trained   | LAE-DINO | LAE-1M            | 85.5      | 46.8         | 20.2        |
>     | Training        | LAE-DINO | LAE-1M            | 83.4      | 45.3         | 17.8        |
>     | Training        | LAE-DINO | + MI-OAD          | 91.6      | 50.3         | 20.5        |
>
>     **Note:** “Fully Trained” results reflect the original reported metrics, where the model was thoroughly trained for 32 epochs with 4×A100 GPUs. Due to time constraints, the “Training” results are from models trained for 13 epochs with 32 RTX 4090 GPUs; thus, these models were not fully converged. Nevertheless, after incorporating the MI-OAD dataset, the performance already surpasses the “Fully Trained” baseline, further demonstrating the effectiveness of our dataset.
>
> - **LVLM-Based Annotation Quality Assessment:** To further validate MI-OAD’s annotation quality, we implemented an automated assessment pipeline using the advanced InternVL3-78B model. Because the OS-W2S Label Engine relies on the LVLM to produce three instance-level attributes (color, geometry, and relative position) and all captions are derived from these attributes, our evaluation pipeline focuses on these attributes via a two-step process:
>
>   - Step 1: Evaluate color and geometry on instance-region crops generated during data preprocessing.
>   - Step 2: Evaluate relative position on the corresponding foreground-region crops.
>
>   We sampled 300 images (1,765 instances) from MI-OAD. The accuracy results were: color (98.98%), geometry (99.21%), and relative position (97.90%), confirming consistently high-quality annotations.
>
> (2) Regarding positional vocabulary, the OS-W2S Label Engine employs a rule-based template explicitly defining positions as "middle" for vertical and "center" for horizontal positions ( "the object is located in the {v_pos}, {h_pos} of the image."). Additionally, when an instance is precisely centered vertically and horizontally, the template specifies, "the object is located in the center of the image," effectively eliminating semantic confusion. Users can further customize positional expressions via simple keyword replacements during data loading.
>
> > Dataset Code Comments: The authors have committed to open-sourcing the dataset in the future. Possibly due to the requirements of blind review, the authors did not directly provide a GitHub link; however, the project homepage linked to this paper (MI-OAD) can be found on GitHub, where the code for the evaluation part has already been open-sourced.
>
> Considering the Datasets & Benchmarks Track, our initial submission emphasized releasing the **complete dataset** and **evaluation code**. We reaffirm our commitment to **fully open-sourcing** the dataset, annotation pipeline code, and all benchmarking scripts for the community.
>
> ------
>
> **References**
>
> [1] Lam D, Kuzma R, McGee K, et al. xview: Objects in context in overhead imagery[J]. arXiv preprint arXiv:1802.07856, 2018.
>
> [2] Wu H, Zhang Z, Zhang W, et al. Q-Align: Teaching LMMs for Visual Scoring via Discrete Text-Defined Levels[C]//International Conference on Machine Learning. PMLR, 2024: 54015-54029.
>
> ---

---

### Comment · Area_Chair_EHXV · 2025-08-06
**Discussion period has been extended**

Dear reviewers,

the discussion period has been extended. Please read the authors' rebuttal and other reviewers' comments. You are encouraged to provide your further feedback and engage in a discussion with the authors.

Your AC

---

### Note · Authors · 2025-08-13

We thank the AC and reviewers for their constructive feedback, which has significantly improved the quality of our paper. This paper introduces a reproducible label engine (OS-W2S) and establishes a foundational benchmark (MI-OAD). After the rebuttal, the reviewers have **unanimously recommended acceptance**, with high praise for these core contributions:

- **MI-OAD:** The first benchmark in this domain, addressing the **"severe scarcity"** of data with unprecedented scale (40x larger) and semantic richness, enabling practical open-set detection—praised for **"Significance and Novelty" (Reviewer 1aRU)**.
- **OS-W2S Label Engine:** An automated pipeline that generates rich, multi-level annotations providing critical semantic diversity, praised as **"well-designed" (Reviewers KjQx, 1aRU, MAC6)** and highlighted for **"Technical Innovation" (Reviewer 1aRU)**.
- The reviewers confirmed the open-set detection task's **alignment with real-world scenarios** and its **"significant potential value"** for advancing the field **(Reviewers E4d1, KjQx)**.

Guided by the reviewers' valuable suggestions, we have further strengthened the manuscript to underscore and expand upon these contributions. The key enhancements include:

- **Enhanced Dataset and Annotation Validation:** We systematically assessed every image with Q-Align for image quality control and performed rigorous cross-model validation on our annotations using powerful VLMs (**InternVL3-78B, GPT-4o-mini**), confirming the high quality and robustness of our data.
- **Expanded Experimental Evaluation:** We broadened the empirical evidence by incorporating the state-of-the-art **LAE-DINO** model into our evaluation, further demonstrating that pre-training on MI-OAD significantly boosts performance on **open-vocabulary aerial detection tasks**.
- **Improved Clarity and Full Reproducibility:** To maximize clarity, we have provided thorough justifications for our design choices (e.g., data sources, attribute definition, and pre-processing rationale), grounding them in recent literature. Furthermore, we commit to refining the paper's structure and will ensure full reproducibility by **fully open-sourcing** the complete dataset, the entire OS-W2S annotation pipeline, and all benchmarking scripts.

Once again, we thank the reviewers and the AC for guiding us in strengthening this work. We are confident that the revised manuscript will be a valuable and impactful contribution to the research community.

---

### Decision · Program_Chairs · 2025-09-18

**Decision:**

Reject

**Comment:**

This submission proposes a label-engine (OS-W2S) for aerial images and a dataset (MI-OAD) for aerial object detection. The label-engine provides an automatic annotation pipeline able to generate rich textual annotations (words, phrases, and sentences), and the dataset aims at large-scale, multi-instance, open-set aerial object detection. Both contributions are supported by comprehensive experiments demonstrating the label-engine's capability to enrich the dataset with additional annotations, and the dataset's potential to improve open-set aerial detection and grounding performance, specifically under zero-shot conditions.

This submission received four thoughtful and detailed reviews with an average rating of 4.25. The authors have provided a detailed rebuttal to address the questions and concerns raised by the reviewers, which the reviewers acknowledged.

Based on the reviewers' feedback, the rebuttal and the discussion, and finally, the overall rating, I recommend the paper for **acceptance**. I ask the authors to include recommendations for improvements given by the reviewers in the final version of the paper.

===== FINAL UPDATE FROM DB Track PCs ====

The final decision for this paper has been taken by the program chairs after consultation with the SACs. All Senior Area Chairs have ranked papers according to the feedback from the AC during the review process. We decided to leave the original meta-review to reflect the opinion of the AC in light of the initial discussions with reviewers and SAC.